# A Computational QSAR, Molecular Docking and In Vitro Cytotoxicity Study of Novel Thiouracil-Based Drugs with Anticancer Activity against Human-DNA Topoisomerase II

**DOI:** 10.3390/ijms231911799

**Published:** 2022-10-05

**Authors:** Doaa M. Khaled, Mohamed E. Elshakre, Mahmoud A. Noamaan, Haider Butt, Marwa M. Abdel Fattah, Dalia A. Gaber

**Affiliations:** 1Histology and Cytology Department, Faculty of Medicine, Helwan University, Cairo 11795, Egypt; 2Chemistry Department, Faculty of Science, Cairo University, Cairo 12613, Egypt; 3Mathematics Department, Faculty of Science, Cairo University, Cairo 12613, Egypt; 4Department of Mechanical Engineering, Khalifa University, Abu Dhabi 127788, United Arab Emirates; 5Histology and Cytology Department, Faculty of Medicine, Misr University for Science & Technology, Cairo P.O. Box 77, Egypt; 6Medical Biochemistry and Molecular Biology Department, Faculty of Medicine, Helwan University, Cairo 11795, Egypt; 7Department of Biomedical Sciences, College of Medicine, Gulf Medical University, Ajman 4184, United Arab Emirates

**Keywords:** QSAR, molecular docking, in vitro cytotoxicity, topoisomerase II, anticancer therapy

## Abstract

Computational chemistry, molecular docking, and drug design approaches, combined with the biochemical evaluation of the antitumor activity of selected derivatives of the thiouracil-based dihydroindeno pyrido pyrimidines against topoisomerase I and II. The IC50 of other cell lines including the normal human lung cell line W138, lung cancer cell line, A549, breast cancer cell line, MCF-7, cervical cancer, HeLa, and liver cancer cell line HepG2 was evaluated using biochemical methods. The global reactivity descriptors and physicochemical parameters were computed, showing good agreement with the Lipinski and Veber’s rules of the drug criteria. The molecular docking study of the ligands with the topoisomerase protein provides the binding sites, binding energies, and deactivation constant for the inhibition pocket. Various biochemical methods were used to evaluate the IC50 of the cell lines. The QSAR model was developed for colorectal cell line HCT as a case study. Four QSAR statistical models were predicted between the IC50 of the colorectal cell line HCT to correlate the anticancer activity and the computed physicochemical and quantum chemical global reactivity descriptors. The predictive power of the models indicates a good correlation between the observed and the predicted activity.

## 1. Introduction

One of the main goals in drug design and drug discovery is to establish a relationship between the biological activity of potential drug-like molecules and the electronic structure and molecular physicochemical properties to provide a means in which to reduce the cost and time of the development of new drugs using mathematical and statistical models yielding the quantitative structure–activity relationship (QSAR). The process of drug development is not straightforward—it is time-consuming, and cost intensive. The process of developing a new drug takes a long time with clinical trials and huge resources and investments. To reduce and minimize the cost and time needed to develop new drugs, computer-aided drug design (CADD) has been proposed to provide this solution. Computational chemistry can compute the electronic structure properties (better known as reactivity descriptors) and physicochemical properties. Biochemical techniques can provide the means to estimate the biological activity. When both the computed descriptors and measured biologically activity are combined, a statistical and mathematical model known as QSAR is produced, which can be considered as the basis for the discovery and design of a new drug.

This study aimed to investigate the structure–activity relationship of selected heterocyclic compounds with promising anticancer activity by linking this biological activity to the computed quantum chemical reactivity descriptors and computed physicochemical parameters by establishing statistical models using multilinear regression (MRL). There is a universal concern regarding finding drugs for curing cancer, which is considered as one of the major causes of mortality worldwide [1]. Chemotherapy is one of the strategies in cancer treatment, where one or several drugs can be used in cancer treatment. Some chemotherapeutic drugs are organic compounds either of natural or synthetic origin. Synthetic heteroaromatic organic compounds that are cyclic compounds with at least two or more different heteroatoms such as N, O, and S have attracted numerous attention due to their wide applications in medicinal chemistry research [2,3,4,5,6,7,8], which among the major pharmaceutical natural products and synthetic drugs have a distinct place because of their remarkable biological activities and established application in medicinal chemistry [4]. The sulfur atoms in these molecules provide great stability in the form of sulfide or disulfide linkages [9].

The novel thiouracil-based indeno pyrido pyrimidines TUIPP, which contain an additional sulfur atom, were chosen for this study, when compared with its analogous, uracil-based indeno pyrido pyrimidine UIPP derivatives [10], which contain only oxygen and nitrogen atoms as heteroatoms based on the proven outstanding biological activity of the latter. The former are expected to exhibit even better antitumor activities than UIPP. TUIPP and UIPP can exist in the dihydro and oxidized forms in the cell tissues, as shown in Figure 1, which shows the oxidation of compound 1H to compound 1O.

Out of the many derivatives of TUIPP, only four selected derivatives were chosen, namely, **2H**, **6H**, **7H**, and **9H** of the dihydro form, which are named from now on as thiouracil-based dihydro indeno pyrido pyrimidine (TUDHIPP), as shown in Figure 2, which were synthesized and characterized by Hassaneen et al. [11]. The potential of the four TUDHIPP derivatives as antitumor agents was explored using density functional theory (DFT) computational approach, molecular docking simulation, virtual screening and drug likeness, and biochemistry techniques to evaluate the antitumor activity evaluated in the form of IC50. The drug design process requires several years for lead identification, optimization [12], and in vitro and in vivo testing [13,14,15,16,17,18] before the first clinical trials [19,20]. A large set of complex data with remarkable uncertainty levels is produced during the process of drug discovery [21].

The discovery and design of new therapeutic chemicals or biochemicals and their development into useful medicine is the science of drug design [3]. The first step in the study of drug design and drug discovery is to perform computational electronic structure calculations. Computational medicinal chemistry has become an important element of modern drug research. Molecular modeling [2] is used, particularly drug design [22,23]. Density functional theory (DFT) is one of the various computational approaches that enable the identification and derivation of many useful and important concepts in chemistry [7,24,25]. In addition, the DFT approach is a useful framework to address chemical reactivity [26] and to compute the electronic structure properties in both the gas and aqueous phases [27]. The next step in drug design and drug discovery is to perform the molecular docking of drug molecules. Molecular docking studies are useful computational and simulation techniques that allow for the assignment of the pocket in which the drug molecules can perfectly fit within the protein. Molecular docking studies can also allow for the assignment of binding sites and the calculation of binding energies between drug molecules and proteins, giving insights into how the drug works inside the cell [28,29]. The third step in the drug design of the molecules is the prediction of their ability to be used as a medicine. There are parameters of critical importance in the development of new drug molecules due to their role in defining overall safety margins, dose intervals, and dose amounts [30], which include absorption-distribution-metabolism-excretion (ADME). The meticulous investigation of physicochemical parameters is essential since medicinal chemists can relate quite easily to such physicochemical parameters [31]. Designing a new drug needs to find the right balance of the physicochemical properties and ADME parameters. Therefore, it is crucial to test whether the proposed derivatives (**2H**, **6H**, **7H**, and **9H**) satisfy the ADME prerequisites as anticancer drugs. The final step in drug design and drug discovery is to perform quantitative structure–activity relationships (QSAR) using the multilinear regression statistical approach that predicts the antitumor activity as a function of the quantum chemical parameters from quantum chemical computation and the physicochemical parameters from virtual screening and drug-like computation [7,24,32,33,34,35,36,37]. In medicinal chemistry, there is a principle indicating that structurally similar molecules may have similar biological activities [38]. The information on the structure of molecules is encoded in molecular descriptors. QSAR models define mathematical relationships between the biological activities of known ligands, and descriptors are used to predict the biological activities of unknown ligands. The QSAR methods have been used to predict and classify biological activities [39,40] of virtual or newly synthesized compounds in various fields including drug discovery [41,42,43].

## 2. Results and Discussion

### 2.1. Methods Benchmark

The choice of computational method and basis set is quite crucial to the accuracy of the calculations. The geometric optimization of the thiouracil (TU) nucleus (Figure 3 was obtained using three different levels of theory. The non-electron correlated ab initio Hartree–Fock (HF) and the electron correlated ab initio/(MP2), where both apply the wavefunction Schrodinger equation and density function theory (DFT), which applies an electron density approach using two different basis sets, namely, 6-311++G(d,p) and cc-pVDZ. The bond lengths from the geometry optimization results are listed in Table 1, which were compared with the bond length results from the X-ray diffraction [44,45] to determine the optimum computational approach that reproduced the experimental x-ray geometrical parameters. The comparison of bond lengths calculated with various methods using the 6-311++G(d,p) and cc-pVDZ basis set of TU is shown in Table 1.

The results show that the density functional B3LYP gives a better agreement with the experimental X-ray data when compared with the ab initio *HF* and *MP2* methods, as indicated by the values of the mean absolute deviation A%, which is calculated as the mean absolute deviations between the calculated and experimental values for each method, as given in the last row of Table 1**.** The results indicate that the B3LYP/6-311++G(d,p) method is the most suitable for predicting the bond length of TU. Therefore, it was applied for the rest of the calculations.

### 2.2. Determination of the Chemical Reactivity of ***2H**, **6H**, **7H***, and ***9H*** Derivatives in the Gas and Aqueous Phases

#### 2.2.1. Effect of Solvent on Energy and Dipole Moment

The effect of solvents on the properties of the molecules was investigated using various theoretical methods [46]. To investigate the effect of solvents on the geometrical parameters of the four derivatives **2H**, **6H**, **7H**, and **9H** in the gas and aqueous phases, the polarizable continuum model (PCM)) solvation model was used. In this model, the solvent is treated as an unstructured continuum outside the solvent-accessible surface of the solute and is characterized only by its dielectric constant, which is 78.5 for water at 25 °C [47]. Table 2 gives the values of the total energy and dipole moments of compounds **2H**, **6H**, **7H**, and **9H**, as shown in Figure 2.

From Table 2, it is obvious that the total energy decreased in the aqueous phase compared to that in the gas phase, making the system more stable in water for all derivatives. For example, for **2H**, the energy difference between the molecule in the gas and aqueous phases was (−0.8 eV), which is related to its solvation energy. The dipole moment values for all molecules increased in the aqueous phase compared to those in the gas phase, indicating that the solvent (water) increases the polarity of the molecules.

#### 2.2.2. Global Reactivity Descriptors

The density functional theory (DFT) defines many important concepts of chemical reactivity using electron density of the drug molecules [48]. These include E_HOMO_, E_LUMO_, energy gap (E_g_), ionization potential (I), electron affinity (A), electronegativity (χ), chemical hardness (η), chemical softness (S), chemical potential (V), electrophilicity (ω), and nucleophilicity (N), which describe the global descriptors used to address the various qualitative concepts in chemical reactivity [46,47], which are calculated at B3LYP/6-311++G (d,p) using the following formulae:V = −χ = −1/2·(I + A),
η = 1/2·(I − A),
S = 1/2·η;
$$\omega =(V^{2}/2\;\eta ),$$
$$N=I_{\left(\mathrm{TCE}\right)}-I$$

Using HOMO and LUMO energies, the vertical ionization potential energy is (I) and the vertical electron affinity is (A), respectively, which can be expressed as I = −E_HOMO_, A = −E_LUMO_, and I_TCE_ is the ionization energy of tetra cyano ethylene. It can be seen that these indices measure the propensity of chemical species toward electrons. Thus, a good nucleophile can be characterized by low values of V and ω, while a good electrophile can be characterized by high values of V and ω [48], where E_HOMO_ characterizes the electron donating ability, while E_LUMO_ characterizes the electron withdrawing ability, which indicate that all molecules are stable [49], as shown in Table 3.

The energy gap (E_gap_) is an important stability index that determines the chemical reactivity of the molecule. From the values reported in Table 3, it was noticed that the energy gap (E_gap_) was small in the aqueous phase for derivatives **2H** and **6H** (which are known to contain electron withdrawing groups), making these molecules more reactive compared to the same molecules in the gas phase. In contrast, compound **9H** (which is known to contain electron donating groups) had a larger E_g_ in the aqueous phase, making them more reactive in the gas phase than in the aqueous phase. The values reported in Table 3 show that the electronic chemical potential (V) and the index of nucleophilicity (N) for compounds **2H** and **6H** (having electron withdrawing groups) decreased in the gas phase compared to those in the aqueous phase. in contrast, compound **9H** (having electron donating groups) had a high N and V in the gas phase compared to those in the aqueous phase. Compounds **2**, and **6H** (electron withdrawing) had lower nucleophilic nature in the aqueous phase than in the gas phase, unlike these compounds, **9H** (electron donating groups) had a higher nucleophilic nature in the aqueous phase than in the gas phase. The electrophilic nature was high in the gas phase than in the aqueous phase for derivatives **2H** and **6H** (electron withdrawing), unlike those in compound **9H** (electron donating), which had a low electrophilic nature in the gas phase than for the aqueous phase. Electronegativity (χ), hardness (η), and softness (S) are reported in Table 3, which are considered as useful concepts for understanding the behavior of chemical systems. Thus, the global hardness (η) reflects the ability of charge transfer inside the molecule, which decreases when the solvent effect is considered, so the molecules **2H** and **6H** were harder in the gas phase than in the aqueous phase. These results may suggest a higher stability of **2H** and **6H** compounds in the aqueous phase than in the gas phase. An opposite trend was found for **9H** compound.

### 2.3. Molecular Docking

#### 2.3.1. Docking with Human DNA Topoisomerase II Alpha (IIα)

Human topo II is the target of several anticancer agents [50], which includes doxorubicin, amascrine, mitoxantrone, and etoposide [51]. Molecular docking of **2H**, **6H**, **7H**, and **9H** of TUDHIPP derivatives with human topoisomerase IIα (4fm9) showed that for the derivatives **2H** and **6H**, a hydrogen bond formation occurred with the active site of topoisomerase IIα, as shown in Figure 4a, and the docking results are tabulated in Table 4.

The results of the molecular docking simulation for compound **2H** from Table 4 and Figure 4a, where four interactions resulting in five hydrogen bonds formed. The first interaction was between the center OE1 of GLN773 and H27N11 of the ligand with a length of 3.11 Å. Another interaction site was between OD1 of ASN770 and H7N1 and H27N11 of the ligand corresponding to a length 2.6, 3.15 Å, respectively. A third interaction site was found to be between OG of SER800 and S8 of the ligand corresponding to a length 3.78 Å. A fourth interaction site was assigned between O3 of DC9.C and O16 of the ligand corresponding to a length 3.58 Å. All interactions resulting in the formation of HBs had a binding energy −9.29 kcal/mol. The results of the molecular docking analysis of compound **6H** from Table 4 and Figure 4a indicates that the formation of five HBs were between compound **6H** and topoisomerase II a. The first was found between centers NE, NH2 of ARG929, and O40 of the ligand with a length of 3.37 and 2.81Å, respectively. The second HB was assigned between OH of TYR892 and O40 of the ligand corresponding to a length of 2.65 Å. The third HB was found between ND2 of ASN779 and H27N11 of the ligand corresponding to a length 3.2 Å. The fourth interaction was found between OP2 of DC10.C and H9N3 of the ligand corresponding to a length of 2.67 Å. The binding energy for all interactions was −9.17 kcal/mol. For compound **9H**, the results in Table 4 indicates that there were two HBs observed in compound **9H**. The first was assigned between OH of GLN773 and S9 of the ligand with a length of 3.12 Å. The second was found between N of GLU854 and O10 of the ligand with a length 3.05 Å. The binding energy of the two assigned HBs was calculated to be −8.53 kcal/mol. In order to verify the relevance of the derivatives **2H**, **6H**, **7H**, and **9H** as potential antitumor agents, a molecular docking study of the well-known antitumor etoposide [51] was performed, as shown in Table 4 and Figure 4b.

Five HBs were observed in etoposide. The first was assigned between NE2 of GLN773 and O1 of etoposide with a length of 2.8 Å. The second was found between O of LYS798 and O9 of etoposide with a length of 3.72 Å. The third was observed between O of SER800 and O9 of etoposide corresponding to a length of 2.81 Å. The fourth was found between O3/DC9.C and O11 of the etoposide corresponding to a length of 3.1 Å. The fifth was assigned between O5/ of DG10.C and O11 of etoposide corresponding to a length of 2.64 Å. The binding energy of the five assigned HBs was calculated to be −8.39 kcal/mol. The comparison between the pattern of binding sites and binding energy of etoposide with topo IIα and those of **2H**, **6H**, **7H**, and **9H** indicate that a good match with **2H** and **6H** was found. This indicates that **2H** and **6H** can be regarded as good candidates as antitumor agents such as etoposide.

#### 2.3.2. Docking with Human DNA Topoisomerase II Beta (II β)

Molecular docking of **2H**, **6H**, **7H**, and **9H** of TUDHIPP derivatives with human topoisomerase II β (3QX3 showed that each compound formed a hydrogen bond with the active site of topoisomerase II β, as shown in Table 5 and Figure 5a.

For compound **2H**, four HBs were observed in compound **2H**. The first interaction of two HBs was assigned between OD1 of ASP479 and H7N1, H27N11 of the ligand with a length of 3.05, 3.13 Å, respectively. The second interaction was found between NZ of LYS456 and S8 of the ligand having a length of 3.17 ÅThe third interaction was found between O3/ of DG10.D and S8 of the ligand corresponding to a length of 3.19 Å. All interactions had a binding energy of −9.29 kcal/mol. In compound **6H**, four HBs were formed. The first interaction of two HBs was found between OD1, N of ASP479 and H7N1, S8 of the ligand with a length of 2.8 and 3.71 Å, respectively. The second interaction was assigned between N1 of DC11.D and O41 of the ligand corresponding to a length 3.81 $A^{0}$. The third interaction was observed between O3/ of DT9.D and S8 of the ligand corresponding to a length 3.26 Å. All HB interactions had a binding energy of −10.07 kcal/mol. To verify the potential relevance of the derivatives **2H**, **6H**, **7H**, and **9H** of TUDHIPP as potential antitumor agents, molecular docking study of the well-known antitumor etoposide [51] with topoisomerase IIβ was carried out and the results are shown in Table 5 and Figure 5b.

The results indicate the formation of three interactions with four HBs in etoposide. The first interaction was assigned between N, OD1 centers of ASP479, and O9 of etoposide with a length of 3.08, 2.78 Å, respectively. The second interaction was found between O3/ of DC8.C and O8 of etoposide having a length of 2.87 Å. The third interaction was observed between O3/ of DT9.D and O9 of etoposide corresponding to a length of 2.81Å. The fourth was found between O3/DC9.C and O11 of etoposide corresponding to a length of 3.37 Å. The binding energy of the four assigned HBs was calculated to be −11.59 kcal/mol. The comparison between the pattern of binding sites and binding energy of etoposide with topoisomerase II β and those of **2H**. **6H**, **7H**, and **9H** indicate an even better match with all derivatives, where the highest one was found in the **6H** compound. This indicates that these four derivatives can be potential antitumor agents such as etoposide.

### 2.4. Drug Likeness and Virtual Screening

Drug-likeness is a useful concept as a guide for molecular drug design in a hit and lead optimization [12,52]. In vivo pharmacokinetic parameters [15,53] such as absorption, distribution, metabolism, and excretion (ADME) are strongly affected by the physicochemical properties of a drug. The analyses of the ADME properties can be achieved using special rules [54]. The parameters of the rules of Lipinski, Veber, Warring and the analysis of the Golden Triangle of the four derivatives **2H**, **6H**, **7H**, and **9H** of TUDHIPP were calculated [54]. The ADME properties that were taken into account include the number of rotatable bonds (n_rotb_), hydrogen bond donors (HBD), hydrogen bond acceptors (HBA), polar surface area (PSA), partition coefficient octanol/water (log P), octanol/buffer (pH 7.4) distribution coefficients (logD_7.4_), and molecular weight (MW). Table 6 shows the results of the physicochemical properties of these molecules.

#### 2.4.1. Lipinski Rule of 5

These parameters indicate the ability of drug molecules toward oral absorption or membrane permeability, which occurs when these molecules follow Lipinski’s rule of five [55]. If these molecules satisfy such a rule, then the chance to be considered as a drug is high. When this rule is applied to the derivatives **2H**, **6H**, **7H**, and **9H**, it imposes restrictions on the partition coefficient of the molecule between water and octanol, the number of hydrogen bond donors (HBD) and hydrogen bond acceptors (HBA), log P, and molecular weight (MW). The Lipinski rule of five states that a molecule can be a potential drug if it satisfies the following: H-bond donors (HBD) with one or more hydrogen atoms is ≤5; H-bond acceptors (nitrogen or oxygen atoms (HBA) is ≤10. The octanol/water partition coefficient log P is ≤5. The molecular weight (MW) is ≤500 Dalton, Da [35]. The index of the number of hydrogen bond donors, HBD (NH and OH), and the number of hydrogen bond acceptors (HBA) (O and N atoms) of the Lipinski rule has been shown to be critical in a drug development procedure, since they have an influence on the absorption and permeation [56]. According to the Lipinski rule, the number of hydrogen bond donors must be less than 5 and the number of hydrogen bond acceptors must be less than 10. The results from Table 6 show that HBD and HBA of **2H**, **6H**, **7H**, and **9H** molecules were found to be within Lipinski’s limit (i.e., less than 5 and 10, respectively).

To predict the solubility of oral drug, Log P is used as an index by partitioning the drug molecule between water and *n*-octanol, the hydrophobic solvent, where the ratio of the concentration of the drug molecule in *n*-octanol and that in water determines the value of P. If it increases, the solubility in water decreases so the absorption decreases. When the results give negative value of Log P, it indicates that the compound is too hydrophilic, hence it has good aqueous solubility, as a result it has better gastric tolerance and efficient elimination through the kidneys. In contrast, when the results give a positive value for log P, it indicates that the drug molecule is too lipophilic, resulting in a good permeability through cell membrane, leading to better binding to plasma proteins, and better elimination by metabolism, the disadvantage being its poor solubility and poor gastric tolerance [57]. The results from Table 6 show that compound **6H** is expected to have the highest hydrophilicity because it has the highest negative log P value, this implies that it will show a good aqueous-solubility, better gastric tolerance, and efficient elimination through the kidneys. The results in Table 6 show that compounds **2H**, **6H**, **7H**, and **9H** had values of log P lower than 5, a requirement by the Lipinski rule to satisfy drug likeness.

There is a relationship between molecular weight (MW) and molecular size, where the larger the molecular size, the larger the cavity formed in water in order to solubilize the compound. An acceptable criterion of the molecular weight of a potential drug molecule is that it should be under 500 Da. Therefore, the smaller the MW of the molecule, the better its absorption. The results from Table 6 show that the MW of all compounds in the chosen series was below the limit (i.e., 500 Da), indicating an easier passage of these molecules through cell membrane. The effect of molecular weight (MW), and hence the molecular size of these molecules, can be conceived as the ability of the drug-like molecule to be soluble in water and lipids (i.e., it should show some degree of solubility in water and lipid phases because as an orally administered drug to reach the inside of a cell, it should be carried in aqueous blood and penetrate the lipid-based cell membrane [50]. In addition, hydrogen bonds play an influential role affecting the water space solubility. For the permeation of potential drug-like molecules into and through the lipid bilayer membrane, these hydrogen bonds need to be broken. Consequently, the increase in the number of hydrogen bonds can result in a corresponding increase in the aqueous solubility, leading to a reduction in the molecule hydrophobicity. Therefore, this may lead to the reduction in partitioning from the aqueous phase into the lipid bilayer membrane for permeation under the influence of passive diffusion. There exists a relationship between molecular size and molecular weight, which shows that the molecular weight is commensurate with the size of the molecule. This means that when the molecular size increases, it may result in the reduction of the compound concentration at the surface of the intestinal epithelium, leading to the reduction in absorption. The increase in molecular size may also result in resisting passive diffusion through the aliphatic side chains of the bilayer membrane [58]. By referring to Table 6, compounds **2H**, **6H**, **7H**, and **9H** satisfy the rules of five of Lipinski, which demonstrates that these derivatives have good absorption and permeation, satisfying the oral bioavailability requirements theoretically. Molecules violating more than one of these indices set by Lipinski can result in problems with bioavailability, and failure to satisfy drug likeness requirements should not be considered for further development as a drug [59].

#### 2.4.2. Veber’s Rule

Veber, et al. [60] provided two additional descriptors to achieve ideal oral bioavailability; the first is the topological polar surface area (TPSA), which should be under 140 Å^2^, and the second descriptor is the number of rotatable bonds (n _rot_), which should be under 10. The rules provided by Veber are based on a solid physicochemical basis, where it is well-established that the larger the number of hydrogen bonds, the lower the solubility in water because these hydrogen bonds should be broken to allow for the permeation of the compounds into and through the cell lipid bilayer membrane [57]. The number of rotatable bonds (n_rot_) measures the molecular flexibility, which is considered to be a good parameter for the oral bioavailability of drugs. (n_rotb_) is defined as any single bond, not in a ring, bound to a non-terminal heavy (i.e., non-hydrogen). Amide C–N bonds are excluded from the count due to their high rotational energy barrier. The topological polar surface area (TPSA) is an important useful parameter for the assessment of drug transport properties. The sum of surfaces of polar atoms (usually oxygen, nitrogen, and attached hydrogen) in a molecule defines the polar surface area. Veber set the limits on TPSA so that molecules with values of 140 Å^2^ or more are expected to have poor intestinal absorption [61]. The results of TPSA from Table 6 of **2H**, **6H**, **7H**, and **9H** of TUDHIPP derivatives were found to be in the range of 70.23–113.37 and were well below 140 Å^2^, which shows that these compounds tend to have an intermediate intestinal absorption. It is obvious that the computed physicochemical parameters tabulated in Table 6 show that all compounds meet the rules set by Lipinski and Veber, which suggests that these compounds theoretically would satisfy the requirements for oral bioavailability.

#### 2.4.3. Golden Triangle

A useful visualization tool of the Golden Triangle was developed by Johnson and co-workers [62], which relates molecular weight (MW) and lipophilicity (Log D at pH 7.4; log D_7.4_). The golden triangle is considered as a container for many different molecular descriptors. This means that compounds that are located inside the Golden Triangle are more likely to be both metabolically stable and to possess good membrane permeability than those outside. Although the Golden Triangle is computed for the nine compounds **1H**–**9H**, its applicability will be demonstrated for the compounds considered **2H**, **6H**, **7H**, and **9H**. The Golden Triangle (Figure 6) shows that compound **9H** is located outside the triangle, indicating poor permeability and clearance. On the other hand, compounds **2H** and **7H** were located inside the triangle, indicating better permeability and clearance [52].

In general, molecules with lower log D and higher molecular weight fail to satisfy drug likeness as a result of their low permeability. On the other hand, molecules with higher log D and higher MW fail to satisfy drug likeness due to elevated vitro clearance [62]. The descriptors of Lipinski and Veber’s rules for etoposide were computed and compared with the compounds **2H**, **6H**, **7H**, and **9H**. The comparison of the computed results with published data on etoposide and the proposed compounds, as given in Table 6, showed a good agreement between the computed Lipinski and Veber’s descriptors of the four derivatives and those published for etoposide, indicating that these molecules can be potential antitumor agents when compared with the well-known antitumor agent etoposide.

#### 2.4.4. Structure–Activity (SAR) Properties

The computed physicochemical properties including molar volume (V), hydration energy (HE), molar refractivity (MR), surface area grid (SAG), and polarizability (Pol) are listed in Table 7 using HyperChem 8.0.7. Molecular polarizability (Pol) is defined as the capacity of the electronic system of a molecule to modulate itself upon the application of an external electric field of light. Molecular polarizability is important in the modeling of many properties including the biological activities of molecules [63]. Molecular polarizability depends only on the molecular volume, where the latter determines the transport characteristics of molecules, which includes blood–brain barrier penetration and intestinal absorption. Therefore, the modeling of molecular properties and biological activity requires the use of molecular volume in SAR studies. The molar refractivity (MR) is another SAR parameter, which is a steric parameter, that depends on the spatial array of the aromatic ring in the molecules. The spatial arrangements are important to study the interaction of the drug molecules with the receptor [64]. Not only does molar refractivity depend on the molecular volume, but also depends on the London dispersive forces that play a strong role in the drug–receptor interaction. The results in Table 7 show that polarizability, molecular refractivity, and surface area grid are generally proportional to the size (Volume) and the molecular weight of the molecules considered. Compound 9H, with the highest molar volume (1070.91 (Å^3^), had the maximum polarizability (45.9 (Å^3^), refractivity (116.3 (Å^3^), and surface area grid (621.57 (Å^3^). Compound 6H, with the molecular volume of 998.3 Å^3^, had a polarizability 42.72 Å^3^, refractivity of 108.18 Å^3^, and surface area grid of 586.26 Å^3^, respectively, but not higher than that of compound **9H**. The hydrophobicity of the drug molecules and its hydration energy were related and the stability of the different molecular conformations in water solutions was determined from the hydration energy [65].

The results in Table 7 show that the increase in hydrophobicity results in a decrease in hydration energy. The change in the values of the hydration energy is affected by the number of hydrogen bond donors and acceptors. Table 7 indicates that compound **6H** has the highest value of hydration energy in absolute value (−16.69 kcal/mol) and is characterized by a high value of hydrogen bond donors (3) and acceptors (6). The ADME properties are largely dependent on lipophilicity. Log P expresses the portioning of the drug molecules between the aqueous medium outside the cell membrane and the lipid nature of the cell membrane. Therefore, compounds with lower Log P are more polar and have poorer lipid bilayer permeability, while compounds with a higher Log P are more nonpolar and have poor aqueous solubility [58]. Compound 6H has a good aqueous solubility and a bad absorption and permeability. Furthermore, Log P values of compounds **2H**, **7H**, and **9H** are in the field of optimal values (0 < Log P < 3 [35]. Therefore, compounds **2H**, **6H**, **7H**, and **9H** have good oral bioavailability and an optimal biological activity. The hydrogen bond interaction of the drug molecules and water play a crucial role in the determination of the relevance of chemical molecules as a drug. When the actual biological environments are considered, where the polar drug molecules are surrounded by water molecules, there is a possibility of hydrogen bond formation between water molecules and these drug molecules. The mechanism of hydrogen bond formation requires the interaction between the proton donor sites of the drug molecule with the oxygen atom of water. A mutual interaction is also possible between the acceptor sites of the drug molecule with the hydrogen atom of water. In Table 7, the hydration energy decreases when the drug molecules contain hydrophobic moieties such as in **7H** and **9H** of TUDHIPP. The presence of hydrophilic groups as in compound (**6H**), shown in Figure 7, having three (HBD): (3 NH) and six (HBA): (four O, 1N and 1S), resulted in the increase in the hydration energy. The SAR descriptors of etoposide were computed as shown in Table 7, which were compared with other published data for etoposide. The results showed a good agreement with the computed Pol and refractivity descriptors. For compound **9H**, the computed polarization and refractivity provided 45.9 Å^3^ and 116.3 Å^3^, respectively, indicating a good agreement with the computed polarization and refractivity of etoposide, giving 55.15 Å^3^, and 138.73 Å^3^, respectively, which suggests that **2H**, **6H**, **7H**, and **9H** can be as effective antitumor drugs as etoposide.

### 2.5. Biological Activity (IC50)

#### 2.5.1. Cytotoxicity Screening of **2H**, **6H**, **7H**, and **9H** TUDHIPP Derivatives

The anti-proliferative activity of the biologically active thiouracil based compounds **2H**, **6H**, **7H**, and **9H** are summarized in Table 8.

Six cell lines were tested including the lung cancer cell line (A549), breast cancer cell line (MCF-7), cervical cancer (HeLa), colorectal cancer (HCT), liver cancer cell line (HepG2), and normal human lung cell line (W138). The **9H** derivative showed the highest cytotoxicity on the colorectal cancer (HCT) cell line (IC50 = 0.06 µM) and human cervical cancer (HeLa) with an IC50 as low as 0.19 µM. It was noticed that the cytotoxicity of **7H** was much less than **9H** on the same selected cancer cell lines except for the HCT cell line, which had moderate cytotoxicity. In contrast, **2H** and **6H** revealed poor cytotoxic effect against the selected cell lines. This may be due to the solubility problem. It is important to note that compound **7H** showed an elevated IC50 (140 µM) toward the normal human lung cell line (W138) compared to an IC50 of 35.09 µM toward the lung cancer cell line (A549), denoting its selectivity to cancer cells and hence the bioavailability and safety of this pro-drug. In a previous study [10], the antiproliferative activity of camptothecin analogues (fused uracil heterocycles) was demonstrated and addressed, where they exhibited cytotoxicity on all of the tested human cancer cell lines at low micromolar concentrations, with the highest potencies toward the LoVo colon, U373 glioblastoma, SKMEL-28 melanoma, and the Jurkat, U373 glioblastoma cell lines.

#### 2.5.2. Visualization of the Morphological Changes in Human Cancer Cell Lines by Phase Contrast Inverted Microscopy

Morphological analysis with phase contrast is one of the best methods to determine apoptosis [66], which is one of the main objectives for cancer therapy [67]. Morphological and biochemical alterations of apoptotic cells include cell shrinkage, nuclear condensation and fragmentation, membrane blebbing [68], and the formation of apoptotic bodies in addition to the loss of attachment to neighboring cells [69]. Biochemical changes in apoptosis include chromosomal DNA cleavage into inter-nucleosomal fragments, the externalization of phosphatidylserine, and the proteolytic cleavage of intracellular substrates [70]. Changes in the HCT and HeLa cell lines were analyzed after 24 and 48 h incubation with **7H** and **9H** using the IC50 doses. Features of activated apoptotic pathways as well as anti-growth effects were noted. Regarding the control (untreated) cells, they maintained their original morphology of having a spindle shaped appearance and most of them were adherent to the tissue culture dish substratum. Nearly 90% confluent adherent cells 24 h after culturing were demonstrated in Figure 8a,c, with the cell confluency relatively increased after 48 h, as shown in Figure 8b,d. Closer observation of the control cells showed vesicular nuclei, prominent nucleoli, and granular cytoplasm, as shown in Figure 9. Few rounded floating cells were observed, which represent those newly divided. Others were rounded and adherent, which represented divided attached cells (Figure 9b,d). The same features were described in previous studies [71,72]. On the other hand, after 24 h incubation of **9H**, the culture cells became less confluent spindle shaped with some rounded or polygonal cells when the concentration of the extract was increased, as shown in Figure 10a,c. A total of 50% confluent spindle shaped cells with few rounded floating cells were recruited (Figure 10d).

A few adherent spindle shaped cells (20% confluent), some polygonal cells and floating unhealthy disrupted cells were demonstrated (Figure 11a,c). Pyknotic bodies of condensed chromatin were shown denoting apoptosis, as shown in Figure 11b,d. In order to analyze the **7H** induced cytotoxicity on the HCT cell lines, both the control and treated cell lines were compared with the control cell line showing spindle shaped attached cells with vesicular nuclei and granular cytoplasm as well as some polygonal cells with multiple processes, as shown in Figure 12. After 24 h incubation, the treated cells showed the same features of apoptotic activity (cell shrinkage, membrane blebbing, nuclear condensation, chromatin cleavage, and the formation of pyknotic bodies of condensed chromatin, as shown in Figure 13a,c. After 48 h of incubation of **7H** and **9H**, more cells became unattached, rounded shaped cells, as shown in Figure 10, echinoid spike and rounded floating cells corresponded to dying or dead cells, and more cellular debris were noted (Figure 13d). In another study [73,74], it has been reported that the cells undergoing apoptosis are manifested in other types of morphological changes such as echinoid spikes on the surface of apoptotic cell, apoptotic bodies, and the decrement in cell number with an increase in the time of incubation. The apoptotic cells produced a loss of cellular adhesion to the substrate and most of the cells even detached from the surface of the tissue culture dish plate and appeared to be floating in the culture medium. Relatively early detachment from their basal membrane is characteristic of the apoptosis of monolayer adherent cells and is called anoikis [74]. It was noted that the result was similar on the HCT cell line after 48 h of incubation with the tested compounds (**7H, 9H**).

#### 2.5.3. Testing DNA Topoisomerase Targeting by **7H** & **9H**

The two compounds with the highest anti-proliferative activity by the MTT assay (**7H** and **9H**) were further tested for their topoisomerase I and II inhibitory activity. Compound **9H** showed an inhibitory effect on topoisomerase I with an IC50 of 38.81 ± 2.27 uM. It also inhibited topoisomerase II, manifested by relaxation of the supercoiled plasmid DNA (form I) at an IC50 of 35.991.79 ± uM, as shown in Figure 14. We can conclude here that these values do not coincide with the anti-proliferative activity of **9H** toward the tested cancer cell lines. Although molecular docking studies suggest that 9H binds with the active site of topoisomerase II, we can conclude here that **9H** may also have another added mechanism for cytotoxicity. The derivative **7H** showed an inhibitory effect on topoisomerase I with an IC50 of 30.92 ± 1.49 uM, and an inhibitory effect on topoisomerase II with an IC50 of 29.49 ± 1.31 uM (Figure 15). It is noteworthy that **7H** inhibitory concentrations toward topoisomerase I and II was almost as its IC50 when tested on in vitro A549, MCF-7, and HeLa cell lines, indicating that this is its mechanism of action as an anticancer compound. Additionally, this denotes its potential safety on normal lung cells in the treatment of lung cancer. In the study conducted by Evdokimov et al. [10], the uracil-fused indenopyridines, prepared as camptothecin analogues, showed the inhibition of topoisomerase II at a concentration range of 0.1–1 µM, but no inhibition of topoisomerase I was revealed by any of the prepared compounds.

#### 2.5.4. Cell Cycle Analysis and Apoptosis

In order to study the effect of TUDHIPPs on the phases of the cell cycle, we analyzed the effect of the IC50 doses of **9H** on the HeLA and HCT cells (cells showing highest cytotoxicity) as well as the effect of the IC50 dose of **7H** on HCT cells by performing a flow cytometry study. Both **9H** and **7H** compounds inhibited the cell growth of HCT and HeLa cells by causing cell cycle arrest in the G2/M phase. A noticeable decrease in the percentage of cells in the G0–G1 phase and S-phase and an increased percentage of cells in the G2/M phase and pre-G1 phase compared to the control HCT cells was detected (Table 9, Figure 16 and Figure 17). Furthermore, **9H** and **7H** markedly increased the percentage of cells undergoing apoptosis compared to the control samples. A noticeable induction of necrosis was also detected (Table 10).

### 2.6. Quantitative Structure–Activity Relationship (QSAR) and Regression Analysis

#### 2.6.1. Quantitative Structure–Activity Relationships (QSARs)

Quantitative structure–activity relationships (QSARs) are attempts to correlate quantum chemical and physicochemical parameters of chemical structures and their biological activity [75,76,77,78,79,80,81,82,83]. QSAR is based on the general principle of medicinal chemistry that the biological activity of a ligand or compound is related to its molecular structure or properties, and structurally similar molecules may have similar biological activities [83]. Such molecular structural information is encoded in molecular descriptors and a QSAR model defines the mathematical relationships between the descriptors and biological activities of known ligands to predict unknown ligand activities. QSAR methods have been applied in several scientific studies including chemistry, biology, toxicology, and drug discovery to predict and classify the biological activities of virtual or newly synthesized compounds [10,11,84,85]. QSAR models can also be used in designing new chemical entities and are regarded as essential tools in pharmaceutical industries to identify promising hits and generate high quality leads in the early stages of drug discovery [86,87,88,89]. Molecular modeling and QSAR calculations are used in many fields [90,91] and drug design [92,93,94,95].

#### 2.6.2. Descriptor Generation

After geometry optimization, molecular descriptors were computed. These are important for the quantitative description of the molecular structure and to find appropriate predictive models [96]. Many descriptors reflect simple molecular properties and can give information about the physical chemical aspects of the activity/property of molecules considered [97]. These properties are related to the intermolecular forces involved in the interaction between ligand and biological receptor and they are also related to the transport and distribution of drugs [98]. It is possible that some of them can give valuable information about the influence of electronic, steric, and hydrophobic features upon the biological activity of the compounds investigated. The computed physicochemical descriptors include molar volume (Vol), hydration energy (HE), partition coefficient octanol/water (logP), the molar refractivity (MR), surface area grid (SAG), molecular weight (MW), molar polarizability (Pol), polar surface area (PSA), and lipophilicity (log D at pH 7.4) using Hyperchem 8.0.7 [99] and Marvin Sketch 18.10.0 [100]. The quantum chemical descriptors: dipole moment (DM), energy of frontier orbital’s E_HOMO_ and E_LUMO_, HOMO and LUMO orbital energy difference, ionization energy (I), electron affinity (A), electronegativity (χ), chemical hardness (η), chemical softness (S), chemical potential (V), electrophilicity(ω) were computed using Gaussian 09 W software [101] using DFT/B3LYP with the 6-311++G(d,p) basis set, as shown in Table 3, Table 6 and Table 7, respectively.

In the present work, the four derivatives **2H**, **6H**, **7H**, and **9H** were evaluated for their growth inhibitory activities against W138**,** A549 MCF-7, HeLa, HCT, and HepG2 (Table 8). In order to determine the role of structural features for QSAR studies, twenty four physical and quantum chemical proprieties of the four derivatives were computed and are described in Table 3, Table 6 and Table 7, respectively.

#### 2.6.3. Regression Analysis

A relationship between the independent and dependent variables (quantum chemical and physicochemical descriptors versus biological activities) were determined statistically using regression analysis. For the composition of the logistic regression model, continuous variables and categorical variables were adopted. The output of results from the stepwise multiple regression model were obtained using MINITAB v. 19 [102]. The physicochemical (pharmacological) and quantum chemical descriptors were used as independent variables and were correlated with the biological activities of the **2H**, **6H**, **7H**, and **9H** derivatives for the generation of QSAR models by multiple linear regression (MLR) analysis. Developing a QSAR model requires a diverse set of data, and thereby, a large number of descriptors have to be considered. Descriptors are numerical values that encode different structural features of the molecules. The correlation between the biological growth inhibition activities (IC50) of HCT and descriptors of the four molecules was performed using four models. Model 1 correlates the antiproliferative activity IC50 and the molecular descriptors. Model 2 correlates the logarithm of the antiproliferative activity (log IC50) and the molecular descriptors. Model 3 correlates the inverse of the antiproliferative activity $\frac{1}{\mathrm{IC}50}$ and the molecular descriptors. Model 4 correlates the inverse of the logarithm of the antiproliferative activity $\log\;\frac{1}{\mathrm{IC}50}$ and the molecular descriptors. In the four models, the values of fraction variance may lie between 0 and 1. The QSAR model having $r^{2}>0.6\;$will only be considered for validation. This will allow us to firmly indicate the correlation between different parameters (independent variables) with anti-proliferative activity against HCT.

For Model 1, the regression equation is computed to give
(1)$$\begin{array}{c} IC50\left(HCT\right)=-223-3.58\;E_{\mathrm{LUMO}}+11.55\;\chi ,\;eV+50.9\;S,\;eV^{-1}\\ -9.089\;\mathrm{logP}-15.61\;n_{\mathrm{rotb}} \end{array}$$
with $r^{2}=0.9682$


In Equation (1), $E_{\mathrm{LUMO}}$, logP, and $n_{\mathrm{rotb}}$ have negative coefficients, while χ and S have positive coefficients. The negative coefficient of $E_{\mathrm{LUMO}}$, suggests that the decrease in the energy of the lowest unoccupied molecular orbital indicates a decrease in the biological activity, IC50. The negative coefficient of log P, which is a measure of the differential solubility of a compound in two immiscible solvents, indicates that the decrease in the solubility (lipophilicity) of the compounds causes an increase in their biological activity. The negative coefficient of the number of rotatable hydrogen bonds, $n_{\mathrm{rotb}}$, indicates that the biological activity increases with the decrease in $n_{\mathrm{rotb}}$.The positive coefficient of the electronegativity χ, which measures the electron attracting power of these molecules, indicates that the biological activity increases with an increase in the electronegativity. Softness (S) is a useful concept for understanding the behavior of chemical systems. A hard molecule has a large energy gap and a soft molecule has a small energy gap. Therefore, soft molecules will be more polarizable than hard molecules. The positive coefficient of the chemical softness indicates that the biological activity of these molecules increases with their polarization. The predictive power of model 1 is demonstrated by plotting the observed activity IC50 vs. the predicted activity, as shown in Figure 18.

For Model 2, the regression equation is
(2)$$\frac{1}{\mathrm{IC}50}\left(\mathrm{HCT}\right)=18.64-6.00\;E_{g},\;\mathrm{eV}-0.289\;\eta +64.6\;S,{\;\mathrm{eV}}^{-1}-5.48\;\mathrm{HBD}$$

In Equation (2), the softness S has positive coefficients, while the energy gap, $E_{g}$, chemical hardness η, and number of hydrogen bond donors (HBD have negative coefficients. This means that $\frac{1}{\mathrm{IC}50}$ increases as the softness S, which measures their polarization, and increases the chemical reactivity of the molecules’ increases. From Equation (3), $\frac{1}{\mathrm{IC}50}$ increases as E_g_ decreases because of the negative coefficient of the energy gap, E_g_, which measures the chemical reactivity of the molecules. η measures the chemical hardness of the molecules. Therefore, a hard molecule has a large energy gap and a soft molecule has a small energy gap. Therefore, hard molecules will be less polarizable than soft molecules. From Equation (3), $\frac{1}{\mathrm{IC}50}$ increases as the hardness of the molecules increases (i.e., as the molecules become less polarizable). The predictive power of Model 2 is demonstrated by plotting the observed activity IC50 vs. the predicted activity, as shown in Figure 19.

For Model 3, the regression equation is
(3)$$\begin{array}{c} \mathrm{Log}\frac{1}{C50}\;\left(\mathrm{HCT}\right)=0.56-0.03488{\;E}_{\mathrm{HOMO}}-0.395\;\chi ,\;\mathrm{eV}+0.213\;V,\;\mathrm{eV}\\ +0.315\;\omega ,\;\mathrm{eV}+1.318\;N,\;\mathrm{eV}+0.1391\;\mathrm{logP}+0.4516{\;n}_{\mathrm{rotb}}\\ +0.634{\;\mathrm{logD}}_{7.4}\; \end{array}$$

In Equation (3), $E_{\mathrm{HOMO}}$ and χ have negative coefficients, indicating that the inverse of the logarithmic of 1/IC50 increases as E_HOMO_ and electronegativity decreases. On the other hand, V, N, log P, n_rotb_, log D _7.4_ have positive coefficients, indicating that the IC50 increases with the increase in the molecular volume, log P, number of rotatable hydrogen bonds, and log D_7.4_. The predictive power of Model 3 is demonstrated by plotting the observed activity IC50 vs. the predicted activity, as shown in Figure 20.

For Model 4, the regression equation is computed to give
(4)$$\begin{array}{cc} \mathrm{LogIC}50\left(\mathrm{HCT}\right)= & -0.56+0.03488{\;E}_{\mathrm{HOMO}}+0.395\;\chi ,\mathrm{eV}-0.315\;\omega ,\;\mathrm{eV}\\ & -1.381\;N,\;\mathrm{eV}-0.1391\;\mathrm{logP}-0.4516{\;n}_{\mathrm{rotb}}-0.634\;\mathrm{logD}\left(7.4\right) \end{array}$$
with $r^{2}=0.9924$

In Equation (3), $E_{\mathrm{HOMO}}$ and χ have positive coefficients, indicating that the logarithmic value of the biological activity, Log IC50, increases with an increase in E_HOMO_ and electronegativity of the molecules. In Equation (2), the electrophilicity ω, nucleophilicity N, logP, $n_{\mathrm{rotb}}$, and $\mathrm{logD}\left(7.4\right)$ have positive coefficients. This means that the biological activity increases as the electrophilicity ω, which measures the stabilization in energy when a system acquires an additional electronic charge from the environment, decreases. The biological activity also increases with the nucleophilicity. The biological activity also increases as logP and $n_{\mathrm{rotb}}$ decreases, Log D is the appropriate descriptor for the lipophilicity of ionizable compounds because it accounts for the pH dependence of a molecule in an aqueous solution, so the negative coefficient of Log D explains that any decrease in the lipophilicity of the molecules causes an increase in biological activity.

## 3. Materials and Methods

### 3.1. Synthesis of ***2H**, **6H**, **7H***, and ***9H***

The four molecules **2H**, **6H**, **7H**, and **9H** were synthesized following the procedure used by Hassaneen et al. [11].

### 3.2. Biological Activity

#### 3.2.1. Testing In Vitro Cytotoxicity (Evaluation of IC50)

The **2H**, **6H**, **7H**, and **9H** compounds were evaluated for their anti-proliferative property using a panel of human cancer cell lines: cervical (HeLA), breast (MCF-7), human liver carcinoma cells (Hepg2), lung cancers (A549), colorectal cancer cells (HCT), and normal human lung cell line (W138). The cells were incubated with different concentrations of the compounds and IC50 was calculated after 48 h by measuring cell viability using the MTT assay [103]. HCT cells were cultured in 75-cm^2^ cell culture flasks using E-MEM supplemented with 10% fetal bovine serum (FBS) as the culture medium. Human breast carcinoma cells (MCF-7) cells, human cervix carcinoma cells (HELA), and human liver carcinoma cells (Hepg2) were cultured using the same conditions except for using the RPMI medium instead of E-MEM medium, and (WI 38) cells with DMEM, human cervix carcinoma cells (HELA) with MEM, and (A549) with RPMI. Growth media were discarded from the cell culture flasks and the cell layer was washed gently with sterile PBS. PBS was decanted, and then the cell monolayer was washed with 5 mL trypsin solution (pre-warmed at 37 °C). Trypsin was decanted and the cell culture flasks were incubated with trace trypsin in the incubator at 37 °C until the cells detached from the surface. Growth media were added to the detached cells. Cells were re-suspended in growth medium to the desired concentration according to cell count. Cell suspension was cultured in another cell culture flask or in 96-well cell culture plates and incubated at 37 °C until confluency [104].

#### 3.2.2. Morphologic Observation Using Inverted Microscope

Morphological changes of the HeLa and HCT cultured cells were observed 24 and 48 h post treatment with **7H** and **9H** compounds. **7H** was used on the HCT cell line only. The control (untreated cells) was also examined with an inverted phase contrast microscope [66].

#### 3.2.3. Topoisomerase I Assay

Topoisomerase I activity was assessed using the relaxation of plasmid DNA as the assay. Topoisomerase I assay kit (cat no 1015-1) was purchased from TopoGEN, Inc., USA. Supercoiled plasmid DNA (form I) was incubated in microfuge tubes with distilled water, buffer, and rising concentrations of the test compounds (5–100 μM) in a final volume of 20 μL. Samples were incubated at 37 °C for 30 min, then terminated with 1/5 volume of the stop buffer. Tubes were placed on ice until ready to load on 1% agarose gel using 1x TAE buffer (50x TAE buffer: 242 g Tris base, 57.1 mL glacial acetic acid, 100 mL 0.5M EDTA). The gel was then stained with 0.5 ug/mL ethidium bromide (15–30 min room temperature), then destained (distilled water) for 10–30 min room temperature. Photographing was conducted using UV trans-illuminator. Trans-illuminated gels were then analyzed by Multi-Analyst software (PC Software for Bio-Rad’s Image Analysis Systems Version 1.1., USA). For each gel, 2 uL of marker (relaxed DNA already in loading buffer) and a reaction without extract to mark the position of supercoiled DNA were loaded.

#### 3.2.4. Topoisomerase II Assay

Topoisomerase II assay kit (cat no TG1001-1) was purchased from TopoGEN, Inc., USA. Briefly, 1 uL of kineotoplast DNA (kDNA) was mixed with 4 uL of 5x Complete Reaction Buffer (made 1:1 of A: B) and 1 uL of test extract and H_2_O to make a final volume of 20 uL. Samples were incubated for 30 min at 37 °C. The reaction was stopped by the addition of 4 uL 5x Stop Buffer, then the samples were loaded onto the agarose gel. The gel was stained with 0.5 ug/mL ethidium bromide and destained for 15 min in water, followed by photographing using a UV transilluminator. A positive control (with known topo II activity) and negative control (no extract) were also used. Topoisomerase II activity was measured by the disappearance of kDNA networks (catenanes) or the formation of decatenated.

#### 3.2.5. Cell Cycle Analysis and Detection of Apoptosis

Cell cycle analysis and the detection of apoptosis was performed using BioVision Annexin V-FITC Apoptosis Detection Kit (Catalog #: K101-25), USA. After inducing apoptosis, cells were collected by centrifugation, re-suspended in 1X binding buffer, then incubated with 5 μL of Annexin V-FITC and 5 μL of propidium iodide (PI) at room temperature for 5 min in the dark. Phosphatidylserine (PS) translocated from the inner face of the plasma membrane of apoptotic cells was detected by staining with a fluorescent conjugate of annexin, then analyzed by flow cytometry. To differentiate between apoptosis and necrosis, PI staining was analyzed by the phycoerythrin emission signal detector.

## 4. Computational Details

### 4.1. Geometry Optimization

All computations were carried out using the Gaussian 09W software package [101]. The molecular geometry of the four compounds was fully optimized using three different methods, Hartree–Fock (HF) [105], Möller–Plesset second order (MP2) level [106], and density functional theory with the Becke’s three parameter exchange functional and the gradient corrected functional of Lee, Yang, and Parr (DFT/B3LYP) [107,108,109,110] and with two basis sets of 6-311++G (d,p) [111] and cc-pvdz [112]. No symmetry constraints were applied during the geometry optimization [106,113]. The vibrational frequencies of each molecule were determined at the same level of theory and it has been checked that all structures correspond to the true minima of the potential energy surface. DFT is used to define the molecular stability and reactivity descriptors to determine a reactive site of the molecule. Dual descriptors are calculated as descriptors of reactivity [107] and optimized geometry visualized using Chemcraft version 1.6 package [114]. The PCM (polarization continuum) model is used for solvation calculation.

### 4.2. Virtual Screening and Drug Likeness

These previously geometry optimized structures were saved for use in the computation of the QSAR parameters from the module QSAR properties in HyperChem software, version 8.0.7 [99]. Using the Calculator Plugins of MarvinSketch 18.10.0 software [100], the calculated parameters of drug-likeness according to the Lipinski rule (the rule of five) [55] and Veber’s rule [60] were obtained.

### 4.3. Molecular Docking Simulation

Molecular docking simulation is performed using four sequential steps: ligand structure preparation, protein structure preparation, ligand–protein docking, and molecular docking analysis. In the ligand structure preparation step, the previously geometry optimized structures of **2H**, **6H**, **7H**, and **9H** of TUDHIPP are saved in the pdb file format. Such Pdb files are converted to the PDBQT file format using Autodock Tools version 1.5.6rc3 [115,116]. In the second step, which involves the preparation of the protein structure for molecular docking simulation, several sequential steps need to be followed. First, it requires the definition of the appropriate protein for docking with **2H**, **6H**, **7H**, and **9H**, which in this case, is the human DNA topoisomerase IIα (PDB ID: 4FM9) [117] and IIβ (PDB ID: 3QX3 [118] were chosen and retrieved from PDB (http://www.rcsb.org/pdb/, accessed on 12 January 2020), followed by molecular mechanics energy minimization using Swiss-Pdb Viewer [119]. Both clean proteins are ready for the next step, which involves the generation of the docking input file using Autodock Tools version 1.5.6rc3. Third, it requires the addition of polar hydrogen atoms, which will be achieved using AutoDock Tools (ADT) 1.5.6rc3, where the Kollman charges are added, AD4 type atoms are assigned, and the protein saved in the PDBQT file format. Fourth, the preparation of the grid and docking parameter files using ADT, where the molecular docking studies are performed using AutoDock 1.5.6rc3 [115,116].

All the bonds of the ligands of **2H**, **6H**, **7H**, and **9H** are rotatable and those of the topo IIα and IIβ are considered as rigid [117]. A grid box size of 90 × 90 × 90 A° with 0.375 A° spacing centered at the site of DNA cleavage of topo–DNA complexes, was selected. In the third step, a molecular docking study was performed for the **2H**, **6H**, **7H**, and **9H** derivatives versus human DNA topoisomerase II, AutoDock1.5.6rc3^®^ suite was used [115,116]. The molecular docking parameters include Lamarckian genetic algorithm (LGAs), mutation rate, and cross over rate.

For the Lamarckian genetic algorithm, an initial population size equal to 150 and the maximum number of evals as 2,500,000 energy evaluations were chosen. A rate of gene mutation of 0.02 and a crossover rate of 0.80 was used. One hundred possible binding conformations were generated using AutoDock1.5.6rc3^®^ tools. Pre-calculated grid maps obtained by Autogrid [116] were used to obtain the docking simulation. In the last step, which involves the molecular docking analysis, the conformers with the lowest binding free energy were selected, after the completion of the docking of the ligand–protein complex using AutoDock1.5.6rc3^®^. These docked low free energy protein–ligand structures were saved as PDBQT files. The van der Waal and hydrogen bond interactions between the **2H**, **6H**, **7H**, and **9H** derivatives and topo II enzyme using the LIGPLOT^+^ program [120] were analyzed. The four docked structures were captured and saved, indicating such interactions.

## 5. Conclusions

Thiouracil-based compounds used in this study are promising potential drugs due to the presence of three hetero atoms. The correlation of their biological activity against HCT and quantum chemical and physicochemical descriptors demonstrates their potential application as antitumor drugs, particularly in treating colorectal cancer. Computational chemistry was used to compute the global reactivity descriptors. Biochemical methods were used to measure the antitumor activity in terms of the IC50 of four thiouracil-based indenopyrido pyrimidines against human topoisomerase of the four cell lines. Morphological analysis using an inverted phase contrast microscope was used to show the apoptotic changes in the HELA and HCT human cancer cells, which further strengthens the findings obtained through the MTT assay. Molecular docking simulation of these four molecules with human topoisomerase computed the binding energy and the binding sites of the pocket of the protein and the drug. A quantitative structure–activity relationship (QSAR) model was established between the IC50 activity and the computed quantum and physicochemical properties of the four molecules. The QSAR equation is quite relevant and shows a potential application for the drug design of this class of molecules as effective anticancer drugs.

## Figures and Tables

**Figure 1 ijms-23-11799-f001:**
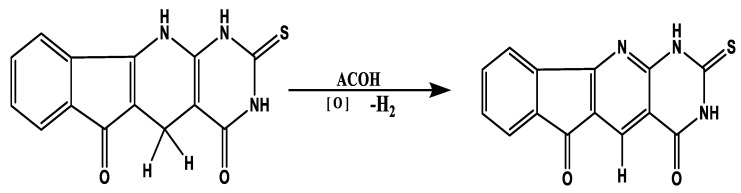
The oxidation of the 1H form of TUDHIPP to 1O.

**Figure 2 ijms-23-11799-f002:**
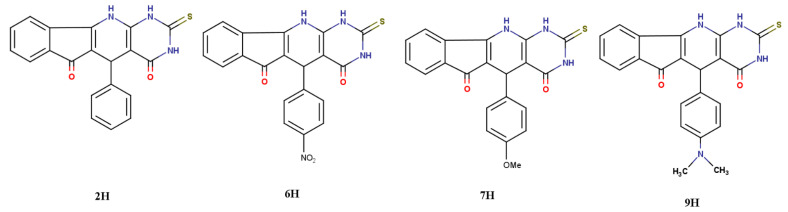
The 2D structure of the four thiouracil based dihydro-indenopyridopyrimidines (TUDHIPP) derivatives (**2H**, **6H**, **7H**, and **9H**).

**Figure 3 ijms-23-11799-f003:**
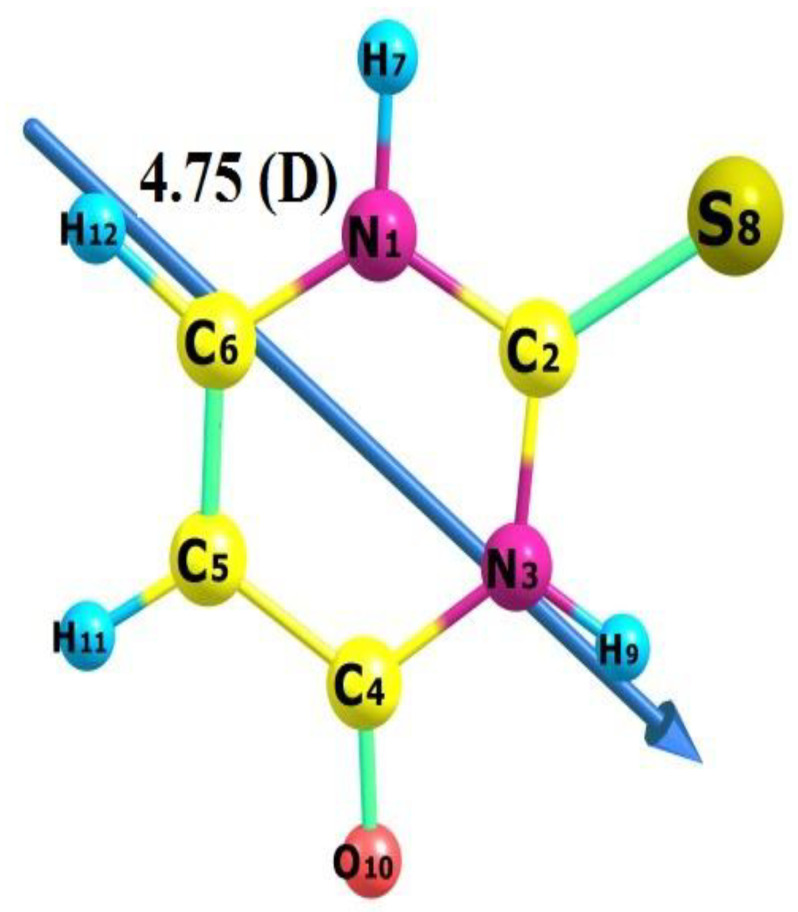
Optimized geometry, numbering system, vector of dipole moment of thiouracil.

**Figure 4 ijms-23-11799-f004:**
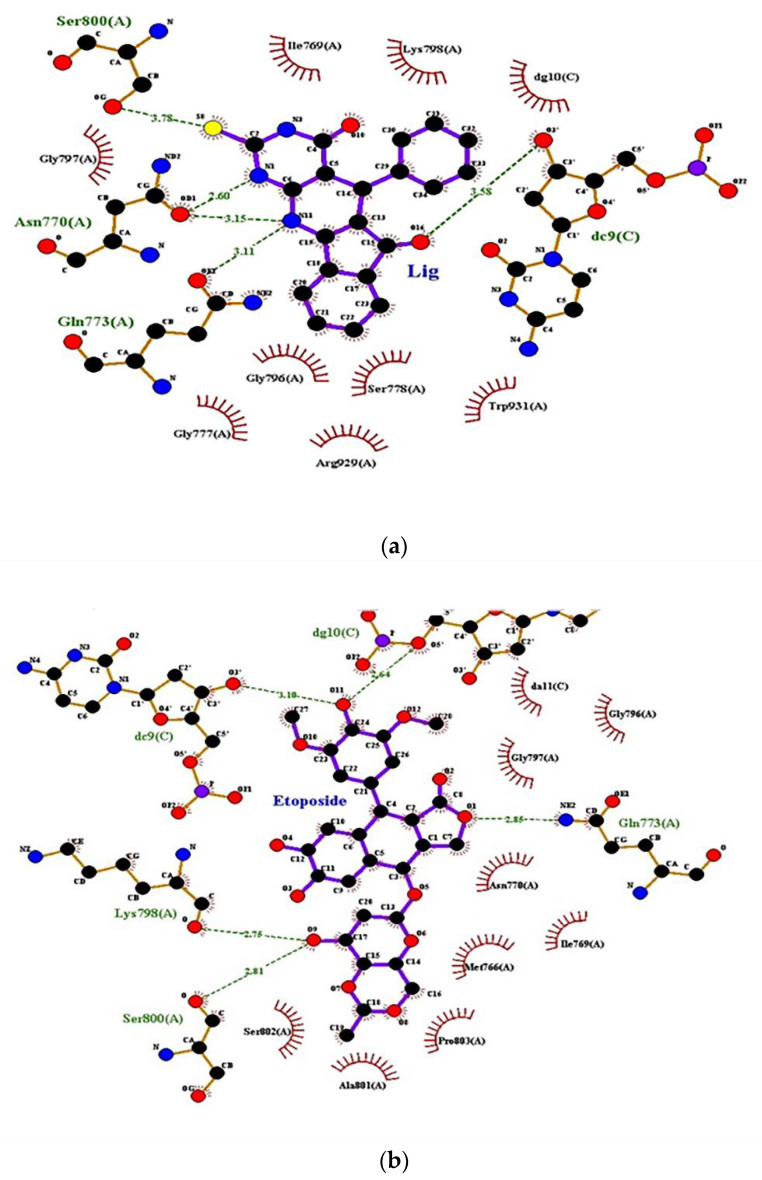
(**a**) Binding sites of the **2H** compound of TUDHIPP with human DNA topoisomerase IIα, 4fm9. (**b**) Binding sites and interaction of etoposide with human DNA topoisomerase IIa, 4fm9.

**Figure 5 ijms-23-11799-f005:**
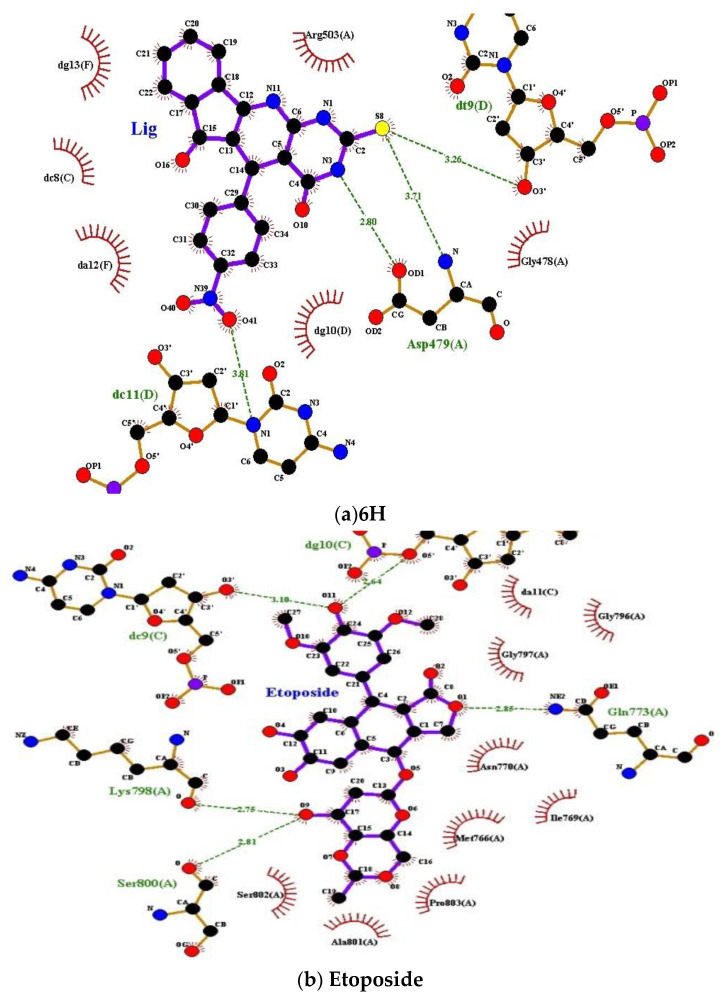
(**a**) Binding sites and interactions of 6H with human DNA topoisomerase IIβ, 3qx3; (**b**) Binding sites and interactions of etoposide with human DNA topoisomerase IIβ, 3qx3.

**Figure 6 ijms-23-11799-f006:**
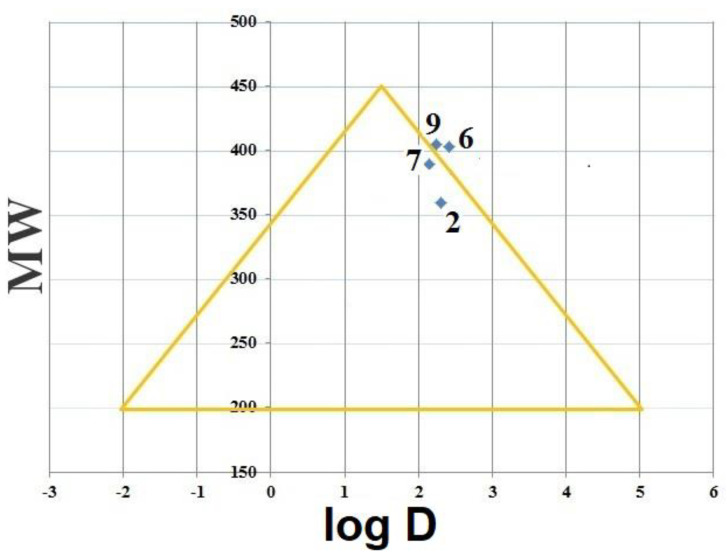
Golden Triangle of DHIPP, compounds **2H**, **6H**, **7H**, and **9H**.

**Figure 7 ijms-23-11799-f007:**
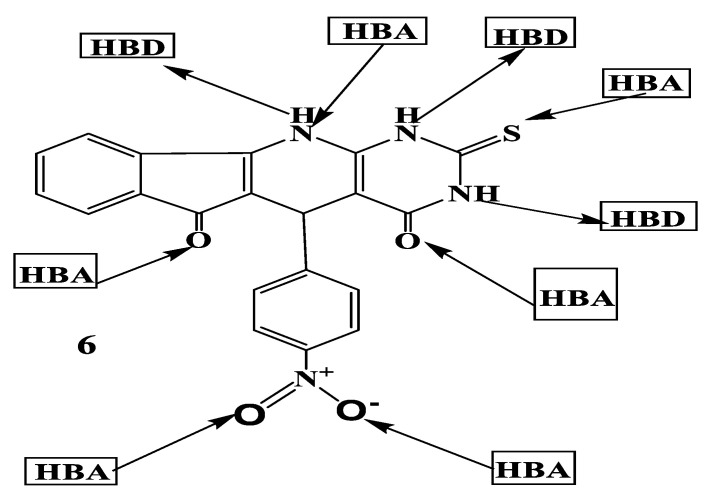
Hydrogen bond donor (HBD) and hydrogen bond acceptor (HBA) sites of compound **6H**.

**Figure 8 ijms-23-11799-f008:**
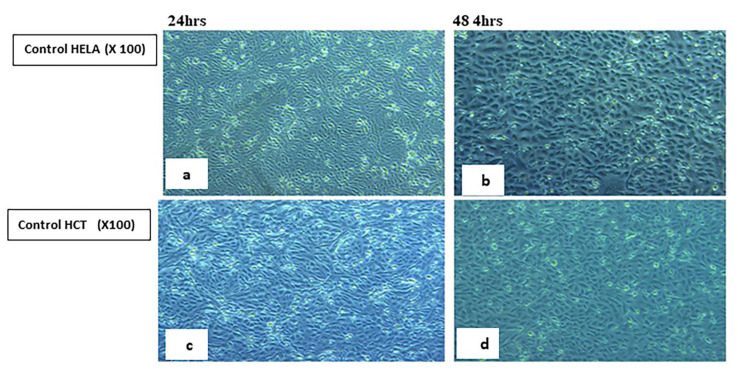
Control (untreated) human HELA and HCT cell lines. Panels (**a**,**c**) show 90% confluent, spindle shaped adherent cells 24 h after culturing. Panels (**b**,**d**) show confluent spindle shaped adherent cells, 48 h post-culture. (Phase contrast, 100×).

**Figure 9 ijms-23-11799-f009:**
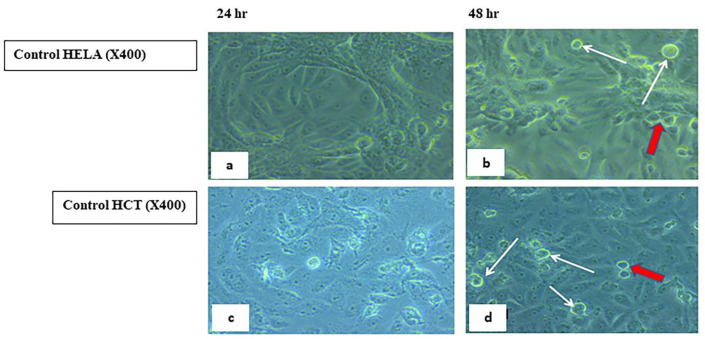
Control (untreated) human HELA &HCT cell lines. Panels (**a**,**c**) show spindle shaped adherent cells with vesicular nuclei and prominent nucleoli. Panels (**b**,**d**) show spindle shaped cells. Notice the rounded floating cells (white arrows) and the dividing just attached cells (red arrows). Phase contrast, 400×.

**Figure 10 ijms-23-11799-f010:**
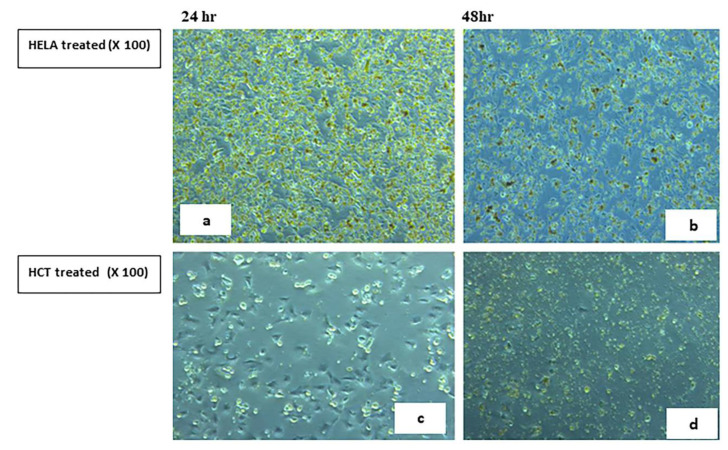
Human HELA and HCT treated cell lines 24, 48 h from incubation with **9H**. Panels (**a**,**c**) show non confluent spindle shaped cells with unhealthy floating cells. Panel (**b**) shows 50% confluent spindle shaped cells with few rounded floating cells with an increase in the concentration of the extract. Panel (**d**) shows numerous cellular debris with few spindle shaped cells compared to (**c**). Phase contrast, 100×.

**Figure 11 ijms-23-11799-f011:**
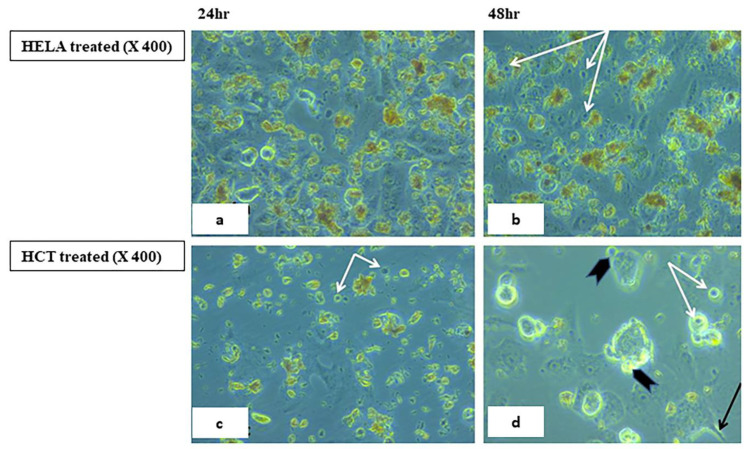
Human HELA and HCT treated cell lines 24,48 h after incubation with **9H**. Panels (**a**,**c**) reveal few adherent spindle shaped cells (20% confluent), some polygonal cells and floating unhealthy disrupted cells. Panel (**b**,**d**, shows few adherent cells with features of apoptosis such as cell shrinkage and pyknotic bodies of condensed chromatin (white arrows), membrane blebbing (arrow heads), echinoid spike (black arrow), and some rounded floating disrupted cells. Phase contrast 400×.

**Figure 12 ijms-23-11799-f012:**
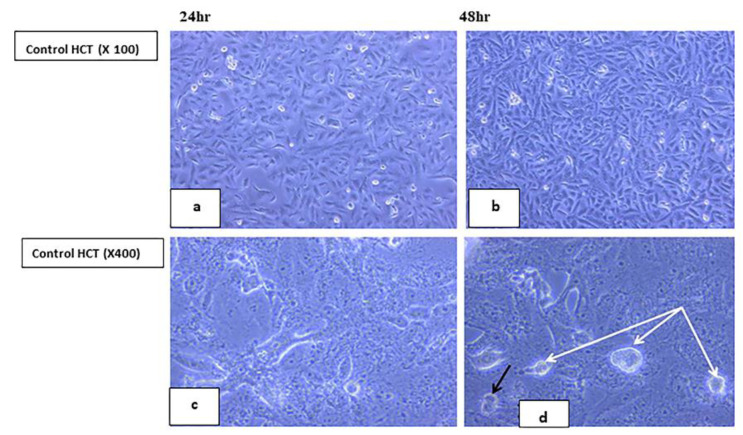
Control (untreated) human HCT cell lines. Panels (**a**,**b**) show spindle shaped attached cells. Panel (**b**) shows 90% confluent spindle shaped cells 48 h post culture. Phase contrast, 100×. Panels (**c**,**d**) show spindle shaped attached cells with vesicular nuclei and granular cytoplasm and also some polygonal cells with multiple processes. Panel (**d**) shows, in addition, rounded floating cells (white arrows) and dividing cells 48 h post culture (black arrow). Phase contrast, 400×.

**Figure 13 ijms-23-11799-f013:**
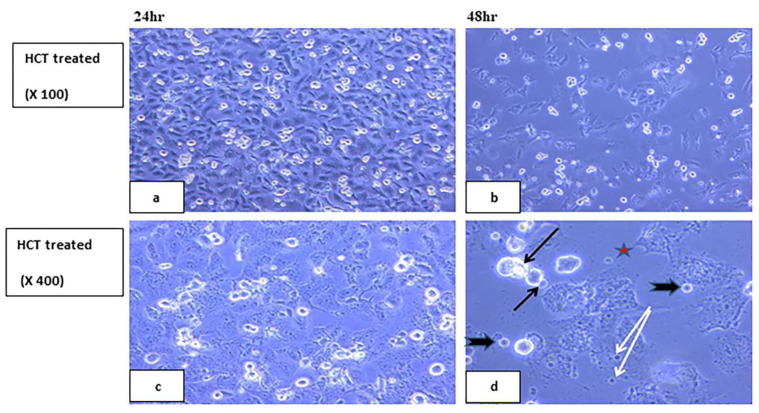
Showing HCT treated human cell lines 24, 48 h after incubation with **7H**. Panel (**a**) shows 80% confluent spindle shaped cells and rounding floating cells. Panel (**b**) shows 40% confluent spindle shaped cells with an increase in the concentration of the extract 48 h post-culture. Phase contrast, 100×. Panel (**c**) shows 70% confluent branched adherent cells with vesicular nuclei, granular cytoplasm, and some rounded floating cells (mostly dead cells). Panel (**d**) shows less confluent polygonal cells, with rounded nuclei and very granular cytoplasm with fewer processes. Notice the rounded floating dead cells and the apoptotic features 48 h post-culture such as cell shrinkage and pyknotic bodies of condensed chromatin (head arrows), membrane blebbing (black arrows), and echinoid spike (star). Phase contrast, 400×.

**Figure 14 ijms-23-11799-f014:**
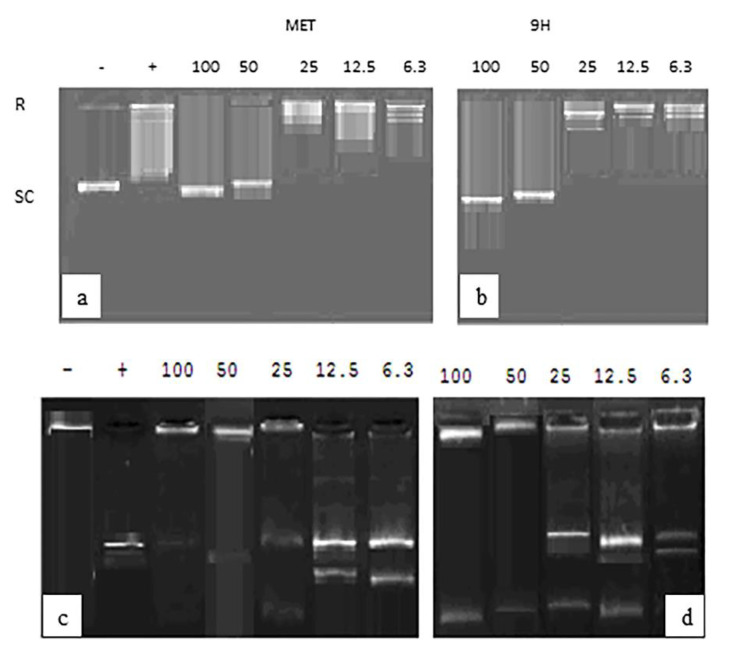
The topoisomearse I and II assay results after incubation with conc 5–100 uM **9H.** (**a**) Supercoiled DNA marker is indicated in lane 1 (-). Lane 2 (+) shows a relaxed DNA marker. Complete activity was seen when no supercoiled DNA substrate remained (last lane) and partial activity with some supercoiled DNA (lane 6) using methotrexate. (**b**) The **9H** compound dose dependent effect on topoisomerase, showing complete inhibition starting from a 50 uM concentration (lane 2) and partial activity at lanes 3 and 4 (25, 12.5 uM, respectively). (**c**) Lane 1: kDNA catenated DNA marker, lane 2, kDNA digested Xho1, Lane 3 to last lane show kDNA + topo II with different concentrations of methotrexate. (**d**) **9H** compound dose dependent effect on topoisomerase II, showing complete inhibition at doses of 100, 50 uM (lanes 1 and 2) and partial inhibition at lane 3 (25 uM).

**Figure 15 ijms-23-11799-f015:**
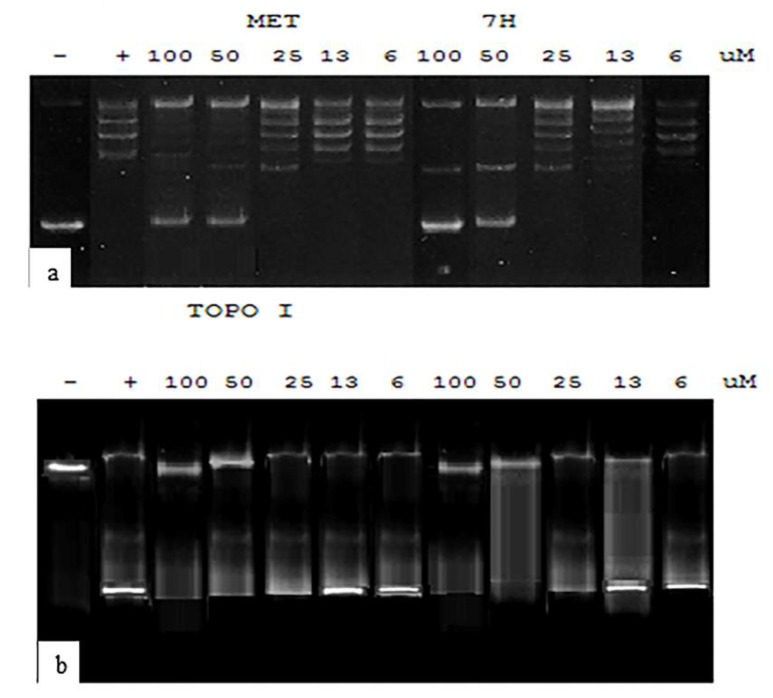
Topoisomearse I and II assay results after incubation with conc 5–100 µM **7H**. (**a**) Supercoiled DNA marker is indicated in lane 1 (−). Lane 2 (+) shows relaxed DNA marker. Complete activity is seen when no supercoiled DNA substrate remains (last lane) and partial activity with some supercoiled DNA (lane 6) using methotrexate. (**b**) **7H** compound dose dependent effect on topoisomerase, showing complete inhibition starting from 50 uM concentration (lane 2) and partial activity at lane 4 (12.5 uM). Lane 1: kDNA catenated DNA marker, lane 2, kDNA digested Xho1, Lane 3 to last lane show kDNA + topo II with different concentrations of methotrexate.

**Figure 16 ijms-23-11799-f016:**
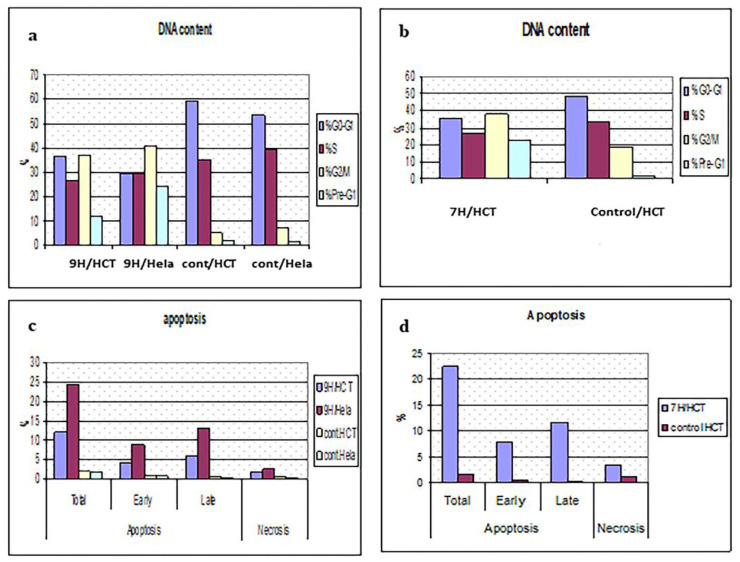
Cell cycle analysis and testing apoptosis. 9H/HCT: Colorectal cells treated with 9H (0.06 µM), 9H/Hela: cervical cancer cells treated with 9H (0.19 µM), Cont. HCT: untreated colorectal cancer control cells, Cont. HeLA: untreated cervical cancer control cells. 7H/HCT: Colorectal cells treated with 7H (1.95 µM). (**a**,**b**) panels reveal that 9H & 7H compounds markedly decrease the percentage of HCT & HeLA cells in G0-G1and S-phase and increase their percentage in G2/M phase, compared to control cells, respectively. (**c**,**d**) panels show that 9H & 7H compounds markedly stimulate apoptosis in HeLA & HCT cells compared to control cells. Also necrosis is induced compared to control cells.

**Figure 17 ijms-23-11799-f017:**
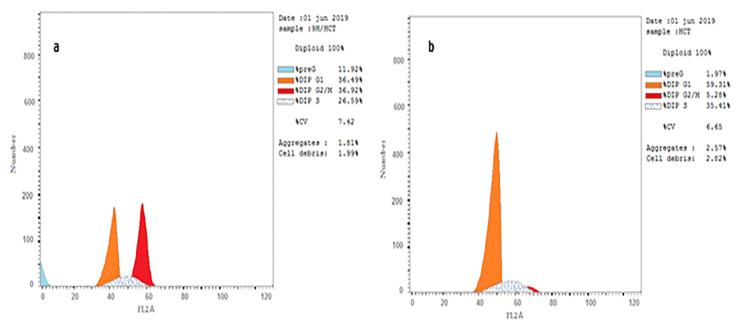
Flow cytometry analysis of cell cycle. (**a**) HCT treated with 9H, (**b**) HCT untreated control. Comparing both curves, it is clear that 9H markedly decreased percentage of cancer cells in G1 phase, thus it has an antiproliferative effect, while it increases the percentage of cells in the G2/M phase, thus it causes cell cycle arrest in G2 phase.

**Figure 18 ijms-23-11799-f018:**
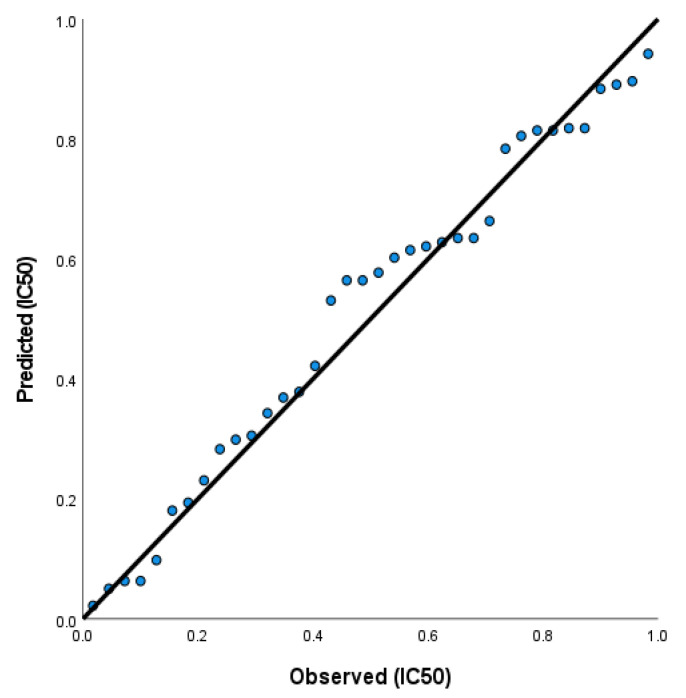
The first QSAR model was obtained by following regression equation $\mathrm{IC}50=-223-3.58\;E_{\mathrm{LUMO}}+11.55\;\chi ,\;\mathrm{eV}+50.9\;S,\;{\mathrm{eV}}^{-1}-9.089\;\mathrm{logP}-15.61\;n_{\mathrm{rotb}}\;$with $r^{2}$, the correlation coefficient, $r^{2}=0.9682$.

**Figure 19 ijms-23-11799-f019:**
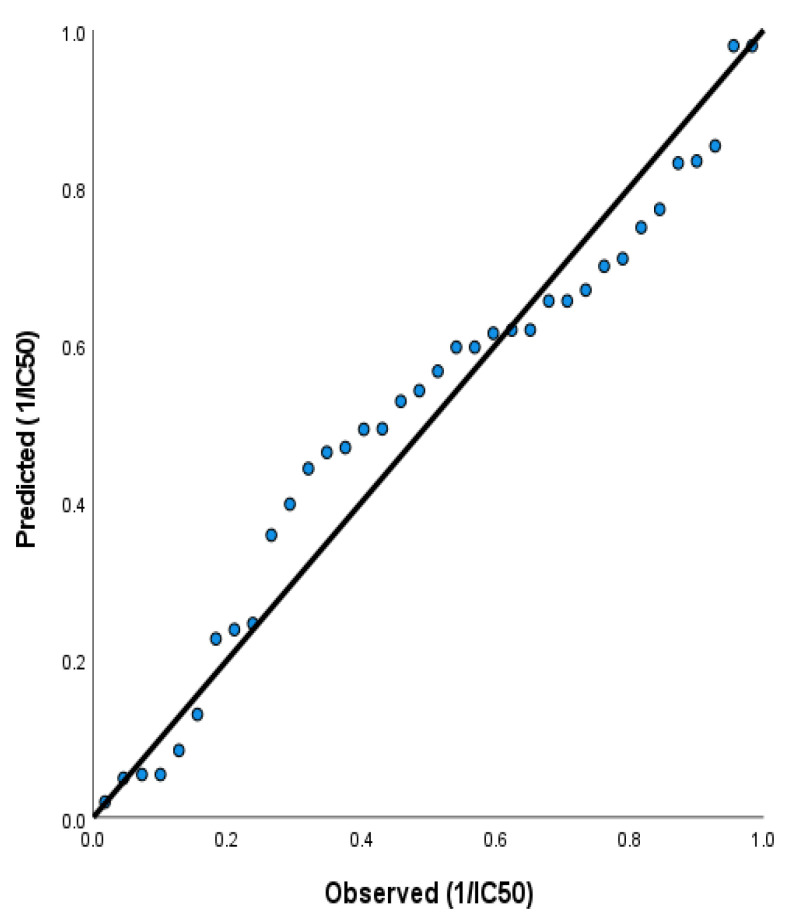
The second QSAR model was obtained by following regression equation $\frac{1}{\mathrm{IC}50}\left(\mathrm{HCT}\right)=18.64-6.00{\;E}_{g},\;\mathrm{eV}-0.289\;\eta +64.6\;S,{\;\mathrm{eV}}^{-1}-5.48\;\mathrm{HBD}$.

**Figure 20 ijms-23-11799-f020:**
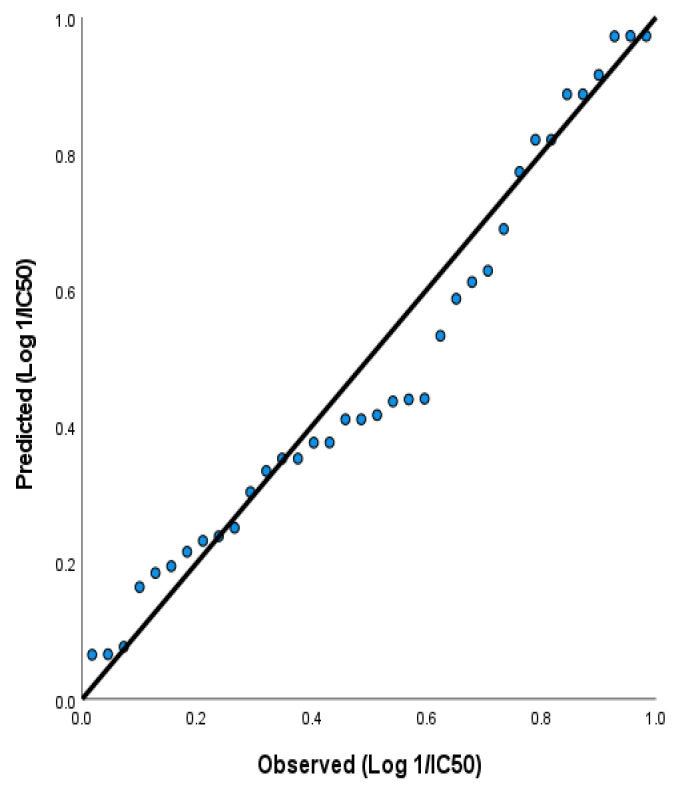
The third QSAR model is obtained by following regression equation $\mathrm{Log}1/\mathrm{IC}50=0.56-0.03488{\;E}_{\mathrm{HOMO}}-0.395\;\chi ,\;\mathrm{eV}+0.213\;V,\;\mathrm{eV}+0.315\;\omega ,\;\mathrm{eV}+1.318\;N,\;\mathrm{eV}+0.1391\;\mathrm{logP}+0.4516{\;n}_{\mathrm{rotb}}+0.634{\;\mathrm{logD}}_{7.4}$.

**Table 1 ijms-23-11799-t001:** The experimental bond length and computed values of the thiouracil (TU) nucleus at different levels of calculations in the gas phase.

Parameters	Exp. *	6-311++G(d,p)			CC-PVDZ		
		ab initio/HF	ab initio/MP2	DFT (B3LYP)	ab initio/HF	ab initio/MP2	(DFT) B3LYP
R(N1,C2)	1.351	1.353	1.380	1.378	1.355	1.382	1.380
R(N1,C6)	1.371	1.374	1.376	1.374	1.374	1.377	1.375
R(N1,H7)	1.030	0.994	1.011	1.010	0.998	1.016	1.014
R(C2,N3)	1.357	1.350	1.376	1.370	1.351	1.376	1.370
R(C2,S8)	1.683	1.661	1.643	1.662	1.665	1.658	1.668
R(N3,C4)	1.394	1.398	1.414	1.417	1.398	1.418	1.419
R(N3,H9)	1.030	0.998	1.015	1.013	1.002	1.021	1.018
R(C4,C5)	1.442	1.460	1.458	1.456	1.462	1.463	1.460
R(C4,O10)	1.235	1.187	1.220	1.214	1.190	1.222	1.217
R(C5,C6)	1.354	1.328	1.356	1.348	1.332	1.362	1.353
R(C5,H11)	1.080	1.070	1.082	1.079	1.077	1.090	1.088
R(C6,H12)	1.080	1.073	1.085	1.083	1.080	1.093	1.091
A%		1.386	1.200	1.102	1.192	1.257	1.105

$A\%=\frac{\mathrm{Mean}\;\mathrm{absolute}\;\mathrm{deviation}\;\mathrm{of}\;\mathrm{bond}\;\mathrm{length}}{\mathrm{Mean}\;\mathrm{bond}\;\mathrm{length}\;\mathrm{of}\;\mathrm{experemential}\;\mathrm{values}\;}\times 100$; ***** Refs. [44,45].

**Table 2 ijms-23-11799-t002:** The total electronic energy au and dipole moment D of the **2H**, **6H**, **7H**, and **9H** TUDHIPP derivatives at the B3LYP/6-311++G(d,p) level of calculation in the gas phase and aqueous phase.

Compound	Gas Phase		Aqueous Phase	
	*E_T_*, au	*μ, D*	*E_T_*, au	*μ, D*
**2H**	−1483.19061	6.96	−1483.22089	11.02
**6H**	−1687.75484	10.68	−1687.79006	15.26
**7H**	−1597.74680	7.78	−1597.77917	12.19
**9H**	−1617.19402	6.59	−1617.22616	10.44

**Table 3 ijms-23-11799-t003:** Global reactivity descriptors of the (**2H**, **6H**, **7H** and **9H**) TUDHIPP derivatives at the B3lyp/6-311++G (d,p) level of calculation in the gas and aqueous phases.

Comp.	Phases	E_HOMO_, au	E_LUMO_, au	E_g_, eV	I, eV	A, eV	χ, eV	η, eV	S, eV^−1^	V, eV	ω, eV	N, eV
**2H**	Gas phase	−0.23738	−0.11034	3.46	6.46	3.00	4.73	1.73	0.29	−4.73	6.47	−2.25
	Aqueous phase	−0.23132	−0.10799	3.36	6.29	2.94	4.62	1.68	0.30	−4.62	6.35	−2.08
**6H**	Gas phase	−0.24851	−0.12131	3.46	6.76	3.30	5.03	1.73	0.29	−5.03	7.31	−2.55
	Aqueous phase	−0.23470	−0.11624	3.22	6.39	3.16	4.77	1.61	0.31	−4.77	7.07	−2.18
**7H**	Gas phase	−0.22399	−0.10882	3.13	6.09	2.96	4.53	1.57	0.32	−4.53	6.54	−1.88
	Aqueous phase	−0.22796	−0.10734	3.28	6.20	2.92	4.56	1.64	0.30	−4.56	6.34	−1.99
**9H**	Aqueous phase	−0.22796	−0.10734	3.28	6.20	2.92	4.56	1.64	0.30	−4.56	6.34	−1.99
	Aqueous phase	−0.20005	−0.10646	2.55	5.44	2.90	4.17	1.27	0.39	−4.17	6.83	−1.23

**Table 4 ijms-23-11799-t004:** Computed molecular interactions of the four derivatives of TUDHIPP (**2H**, **6H**, **7H**, and **9H**) with human DNA topoisomerase II α (4fm9).

Ligand	Run No.	Interaction Residue in Human Topo II α	Interaction Atoms (Amino Acid…Ligand) HB Length (Å)	H Bonds Formed	Binding Energy kcal/mol	Inhibition Constant K_i_, nM
**2H**	18	GLN773.A ASN770.A SER800.A DC9.C	OE1….27N11 (3.11) OD1….H7N1 (2.6), OD1….H27N11(3.15) OG….S8 (3.78) O3….O16 (3.58)	5	−9.29	155.7
**6H**	3	DG10.C ARG929.A TYR892.A ASN779.A	OP2….N3H9 (2.67) NE….O40 (3.37), NH2….O40 (2.81) OH….O40 (2.65) ND2….H27N11 (3.2)	5	−9.17	189.18
**7H**	48	SER778.A GLU854.A TYR892.A ARG929.A ASN779.A	OG….H7N1 (3.18) N….O16 (2.99) OH….O39 (2.69) NH2….O10 (3.17) OD1….H27N11 (2.6)	5	−9.23	170.11
**9H**	93	TYR892.A GLU854.A	OH….S8 (3.121) N….O10 (3.05)	2	−8.53	555.3
**Etoposide**	43	GLN773.A LYS798.A SER800.A DC9.C DG10.C	NE2….O1 (2.85) O….O9 (2.75) O….O9 (2.81) O3^/^….O11 (3.1) O5^/^….O11 (2.64)	5	−8.39	708.84

**Table 5 ijms-23-11799-t005:** Computed molecular interactions of the four derivatives of TUDHIPP (**2H**, **6H**, **7H**, and **9H**) with human DNA topoisomerase II *β* (3qx3).

Ligand	Run No.	Interaction Residue in Human Topo II β	Interaction Atoms (Amino Acid…Ligand) HB Length (Å)	H Bonds Formed	Binding Energy kcal/mol	Inhibition Constant K_i_, nM
**2H**	4	ASP479 LYS456 DG10.D	OD1….H7N1 (3.05), OD1….H27N11 (3.13) NZ….S8 (3.17) O3/….S8 (3.19)	4	−9.29	154.3
**6H**	28	ASP479 DC11.D DT9.D	OD1….H9N3 (2.8), N….S8 (3.71) N1….O41 (3.81) O3/….S8 (3.26)	4	−10.07	41.62
**7H**	75	GLN778.A DC8.C	N….O10 (3.51) O3/….S8 (3.58)	2	−9.34	142.45
**9H**	49	ASP479.A DT9.D	OD1….H7N1 (3.27), OD1….H27N11 (3.04) O3/….H27N11 (3.42)	3	−9.16	193.43
**Etoposide**	19	ASP479.A DC8.C DT9.D	N….O9 (3.08), OD1….O9 (2.78) O3/….O8 (2.87) O3/….O9 (3.37)	4	−11.59	3.17

**Table 6 ijms-23-11799-t006:** Computed pharmacological activities and properties using multi parameter optimization (MPO) method for the four TUDHIPP derivatives (**2H**, **6H**, **7H**, and **9H**).

Compounds	Lipinski Rules					Veber Rules		Log D(7.4)
	Log P	MW (DA)	HBD	HBA	Lipinski Score of 5	n_rotb_	PSA	
**2H**	1.39	359.4	3	4	4	1	70.23	2.310
**6H**	−2.52	404.4	3	6	4	2	113.37	2.250
**7H**	1.14	389.43	3	5	4	2	79.46	2.152
**9H**	1.66	402.47	3	5	4	2	73.47	2.416
**Etoposide**	1.27	588.56	3	13	2	5	160.83	1.15
**Etoposide ***	1.16	588.56	3	13	2	5	160.83	
**Etoposide #**		588.56	3	13	2			0.74

* https://www.drugbank.ca/drugs/DB00773, accessed on 12 January 2020; # *Chem. Pharm. Bull.* 61(12) 1228–1238 (2013).

**Table 7 ijms-23-11799-t007:** The SAR properties of **2H**, **6H**, **7H**, and **9H** derivatives of TUDIHPP.

Comp.	Polarizability (A^3^)	Refractivity (A^3^)	Vol (A^3^)	Surface area (Grid) A^2^	HE (kcal/mol)
**2H**	40.88	101.87	739.85	554.14	−11.73
**6H**	42.72	108.18	998.3	586.26	−16.69
**7H**	43.35	108.33	1014.72	598.73	−13.42
**9H**	45.9	116.3	1070.91	621.57	−10.22
**Etoposide**	55.15	138.73	1448.08	808.26	−25.39
**Etoposide ***	58.77	139.02			
**Etoposide #+†**	55.5	140.1			

* https://www.drugbank.ca/drugs/DB00773, accessed on 12 January 2020; + https://smfmnewsroom.org/elements/etoposide-c29h32o13-structure/, accessed on 12 January 2020; # https://www.lookchem.com/Etoposide/, accessed on 12 January 2020; † https://www.lookchem.com/Etoposide/, accessed on 12 January 2020.

**Table 8 ijms-23-11799-t008:** The antiproliferative activity (IC50 in µM) of thiouracil-based camptothecin analogues.

		W138	A549	MCF-7	HeLa	HCT	HepG2

**2H**		---	---	---	---	22.80	---
**6H**		---	---	---	---	43.25	---
**7H**		140.48	35.09	24.08	30.42	1.95	9.98
**9H**		0.34	0.28	0.32	0.19	0.06	1.112

W138: normal human lung cell line, A549: lung cancer cell line, MCF-7: breast cancer cell line, HeLa: cervical cancer, HCT: colorectal cancer, HepG2: liver cancer cell line.

**Table 9 ijms-23-11799-t009:** Cell cycle analysis after incubation with **7H** and **9H**.

DNA Content				
Code	%G0–G1	%S	%G2/M	%Pre-G1
Control HCT	59.31	35.41	5.28	1.97
**7H**/HCT	35.77	26.38	37.85	22.42
**9H**/HCT	36.49	26.59	36.92	11.92
Control HeLA	53.48	39.28	7.24	1.63
**9H**/HeLA	29.64	29.57	40.79	24.36

Control HCT: untreated colorectal cancer control cells, **7H**/HCT: colorectal cells treated with **7H** (1.95 µM), **9H**/HCT: colorectal cells treated with **9H** (0.06 µM), control HeLA: untreated cervical cancer control cells, **9H**/Hela: cervical cancer cells treated with **9H** (0.19 µM).

**Table 10 ijms-23-11799-t010:** The apoptosis and necrosis pattern in the control and treated cancer cells treated with **7H** and **9H**.

		Apoptosis			Necrosis
		Total	Early	Late	
Control HCT		1.97	0.79	0.63	0.55
**7H**/HCT		22.42	7.59	11.56	3.27
**9H**/HCT		11.92	4.26	5.94	1.72
Control HeLa		1.63	0.89	0.35	0.39
**9H**/Hela		24.36	8.72	12.95	2.69

## Data Availability

Not applicable.

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
