# Peer review of "A Computational QSAR, Molecular Docking and In Vitro Cytotoxicity Study of Novel Thiouracil-Based Drugs with Anticancer Activity against Human-DNA Topoisomerase II"

_ijms, 2022, doi:10.3390/ijms231911799_

Round 1

Reviewer 1 Report

1. Line 41 in introduction needs reference.

2. Line 49 in introduction needs reference.

3. Lines 82-82 has to be at the beginning of the introduction.

4. Figure 16, error bars are needed.

5. Docking validation is missing

6. In conclusions you have to mention some comments about your thiouracil-based drugs. (1-2) lines. Include the values that made your results important.

Best regards,

Author Response

1- in line 41, the statement doesn't need reference

2- in line 49, the statement doesn't need reference

3- line 82 has been moved to the beginning of the introduction 

4- in figure 16 (cell cycle analysis and testing apoptosis ,there is no need for error bars. it is not essential and the results are quite indicative without error bars.

5- The purpose of molecular docking simulation is to predict the protein pocket in which the binding site of the drug with the protein are interacting and computing the binding energies and deactivation constant which are given in table 4 &5.

 6- Some comments  on thiouracil-based drugs are included in the conclusion section as suggested by the reviewer. 

Reviewer 2 Report

The manuscript by Khaled et al. describe in silico investigation of several thiouracil-based compounds along with the experimental data on their biological activities. The quantum chemical (HF, MP2, DFT) and molecular docking methodologies were applied. ADME descriptors were also calculated. The experimental data were related to the theoretical descriptors to obtain QSAR models. The results are important and can be useful for the development of new anti-cancer drugs.

However, the manuscript has many drawbacks listed below. 

Line 59:  The phrase "different atoms heteroatoms" should be replaced with "different heteroatoms".

Figure 1:  "ACOH" above the arrow should be replaced with "AcOH". Also, "-H2" has to be indicated only if molecular hydrogen is formed. Otherwise, just leave the schematic symbol for an oxidant under the arrow.

Line 128:  The basis set notation should be written without spaces: "6-311++G(d,p). Please check also throughout the manuscript.

Table 1:  It is not clear whether the computed molecular geometries were optimized in gas phase or with a solvation model. Please indicate.

Table 2:  Please use the unique notation for the functional throughout the manuscript: "B3LYP", i.e. in capital letters. The same refers to "Water phase" which should be replaced with "Aqueous phase" or "Aqueous solution".

Figure 2:  Quality of the figure is low and size of symbols in the structural formulae is very small.

Line 172:  Please replace "corresponds to its solvation energy" with a more cautious phrase "is related to its solvation energy".

Lines 196-198:  The last sentence of the paragraph should be removed, as the negativity of frontier orbital energies actually do not give full evidence for stability of the molecules.

Table 3:  The numbers for compound 7H are shifted to the right. Please correct.

Tables 4 and 5:  It should be clearly indicated if all the data were obtained by molecular docking. For example, it is not clear whether Ki values are experimental or calculated.

Figures 4-5 have different horizontal and vertical scales. Please adjust the aspect ratio. Also, the figure numbering as 4a, 4b, 5a, and 5b is not acceptable. Please use a one-digit numbering or place two panels (a and b) in the same figure.

Line 255:  The angstrom symbol is incorrect. Please check throughout the manuscript (should be Å).

Line 349:  Regarding the Golden Triangle, a corresponding literature reference should be given.

Table 6:  Again, it should be clearly indicated which of the parameters were calculated, and which of them were experimentally determined.

Sections 2.2.1 and 2.2.2 contain too detailed and trivial definitions of ADME parameters. These sections should be shortened. The same refers to Section 2.6.1.

Figure 16 should be improved regarding resolution (Panel d), font size and aspect ratio (panels a-c).

Figure 17 contains too small text.

Line 893:  It is not clear why the authors correlated both log(IC50) and log(1/IC50) as dependent variables in different QSAR models. Actually these are mutually equivalent variables:  log(1/IC50) = -log(IC50).

Line 903:  The r^2 value of 96.82 is reported. This is incorrect, as r^2 cannot exceed 1.0. Perhaps, the author meant 0.9682? The same refers to Equation 4.

Equations 1-4:  Along with r2, other characteristics of QSAR models should be reported: dispersion or standard deviation and/or F-statistics.

For each of the derived QSAR models, the authors write a sentence concerning predictive power of the model, without any explanation of how the predicted values were obtained. If these values are just calculated dependent QSAR variable at each data point, then this is fitting, not prediction. The predicted power can be evaluated via, for example, leave-one-out or leave-N-out procedures.

Figures 18-20:  If the vertical axes show calculated (not predicted) values, then the corresponding text at the axes should be changed.

Section 3.3.1:  The solvation model used in quantum chemical calculations should be mentioned here.

Line 1071:  "cleaning both proteins from heteroatoms" is an inappropriate phrase. Please modify.

Lines 1080-1081 contain a strange special symbol (looks like a spiral). Please clarify.

Among numerous literature citations, there are no papers published after 2017. At the same time, papers published between 2010 and 2015 are actively cited. In my opinion, references to the relevant literature published in the last five years should also be included in the manuscript.

Polishing of English language of the manuscript is necessary.

Summarizing, I recommend major revision of the manuscript before acceptance.

Author Response

1- In line 61 not line 59 (as suggested by the reviewer), the different atoms heteroatoms  are replaced by the statement different heteroatoms (indicated as red colour.

2- In figure 1, we leave it the way it is because removal of H2 is equivalent to oxidation. We indicate oxidation by including the letter (O).

3-Line 128:  We keep basis set notation as "6-311++G(d,p) without spacing along the text as the reviewer  suggested.

4- In table 1, we indicate that the calculations are the Geometry thiouracil base is performed in the gas phase and typed in red.

5- In table 2, the functional B3LYP is rewritten as capital letters and water phase is changed to Aqueous phase and typed in red as the reviewer  suggested.

6- The reviewer is right but this is the best we can supply at the present time.

7- The  phrase  "corresponds to its solvation energy"  is replaced by  phrase "is related to its solvation energy and typed red as the reviewer  suggested.

8- I agree with the reviewer The last sentence  between line 196-198 is removed as the reviewer  suggested.

9-  Regarding table 3, The reviewer  is right , we will correct the numbers shifted for compound 7H.

10- In table 4 & 5, the molecular docking data are computed and not experimental determined and typed in red color as the reviewer  suggested.

 11- In fig 4& 5 there are no scales used . it is a scaleless figures indicating the pocket in the protein where interaction with the drug takes place.

12-  The angstrom sign in line 255 is corrected in red.

13- Line 349, regarding the Golden Triangle reference 83 is referring to Golden Triangle as well.

14- In table 6, all physicochemical parameters are computed as indicated by the red color table caption.

15- Sections 2.2.1 and 2.2.2 2.6.1   it will be misleading if shorten that's why we keep it in its present form.

16- Regarding  Figure 16 , we do our best, it will be improved and reuploaded.   .

17- Regarding  Figure 17, this the best form we can get at the present time. it will be improved and reuploaded.   .

18- Regarding line 893, the authors try to establish all the possibilities of the QSAR models. 

19- Regarding line 903, the reviewer  is right and we corrected the     r2  value  

20- For equation 1-4, this is the best we can do for the statistical  parameters of the QSAR model. regarding the software used 

21- We agree with the reviewer that it is fitting.

22- The vertical axes are predicted.

23- The PCM (polarization continuum) model is used for solvation calculation as typed in red.

24- We deleted the phrase from the text.

25-  regarding Lines 1080-1081, strange symbols  has been corrected to IIα and IIβ as typed red in the text.

26- To the best of our knowledge there are no literature on thiouracil base drugs addressing QSAR and biological activity at the time of writing manuscript and this is the best literature offered.

27- Some sentences were added a to results and discussion (point 2.5.2) and  typed in red.

28- Regarding material & methods, one sentence was  added  to point 3.2.2 and typed in red.

29- Regarding the conclusion, one sentence was added from the begining of morphological changes  and typed in red 

28- References 50, 93 &101 were updated and typed in red

Reviewer 3 Report

This paper aims to provide the investigation of the structure-activity relationship of selected heterocyclic compounds with promising anticancer activity by linking this biological activity to computed quantum chemical reactivity descriptors and computed physicochemical parameters by establishing statistical models using Multilinear Regression (MRL).

The manuscript is written comprehensively enough to be understandable despite of the complexity of the subject.

They addressed this aim by establishing a quantitative-structure activity relationship (QSAR) model between the IC50 activity and the computed quantum and physicochemical properties of the four molecules.

The paper stated the purpose, discussion and global implication are clearly stated and consistent with the rest of the manuscript; authors provided the required test and analysis and enough information in their discussion by using a good number of important articles talked about the subject.

The authors clearly elucidated that they used various biochemical methods to evaluate the IC50 of the cell lines and the QSAR model is developed for colorectal cell line HCT as a case study.

The authors addressed their hypothesis and opinion in a reproducible way and proved their results through all the required experiments and analysis and they used enough number of analyses to prove their results. The results were presented in a clear way which facilitate in reaching a conclusion.

The abbreviations should be explained at the first place they are mentioned.

In vitro, in vivo, et al.: should be written in italic.

The figures are not clear enough, authors should use a better resolution for the images they used.

Authors should follow the journal guidelines when they draw the Molecules’ structures, they should use the right Chemdraw settings.

No plagiarism has been detected.

References: The authors perfectly followed the journal guidelines.

Author Response

All the comments given by reviewer 3 are taken into account when considering the comments by  reviewer 1 &2  we would like to thank him for critical reading the manuscript . The in vivo, in vitro & et al will be corrected to be  italic along the manuscript 

Round 2

Reviewer 1 Report

ok

Author Response

Thank you for your time and effort.

Reviewer 2 Report

The authors have substantially improved the manuscript. Now it can be accepted for publication. Nevertheless, I recommend improving the quality of Figure 2 containing the structural formulae of the compounds. Although the authors declare that "this is the best we can supply at the present time", it is obvious that ordinary chemical graphics software (ChemOffice, ISIS Draw, Marvin Sketch, and others) could provide a much better quality of the drawing. 

Author Response

Thank you for your time & effort. Figures 2&6 were modified and clarified with much better resolution.

All responses were marked with track changes.